# Inhibitory gating of coincidence-dependent sensory binding in secondary auditory cortex

Amber M. Kline [1,2], Destinee A. Aponte[1,2], Hiroaki Tsukano[1,2], Andrea Giovannucci [2,3] &
Hiroyuki K. Kato [1,2,4✉]

Integration of multi-frequency sounds into a unified perceptual object is critical for recognizing syllables in speech. This "feature binding" relies on the precise synchrony of each component's onset timing, but little is known regarding its neural correlates. We find that multi-frequency sounds prevalent in vocalizations, specifically harmonics, preferentially activate the mouse secondary auditory cortex (A2), whose response deteriorates with shifts in component onset timings. The temporal window for harmonics integration in A2 was broadened by inactivation of somatostatin-expressing interneurons (SOM cells), but not parvalbumin-expressing interneurons (PV cells). Importantly, A2 has functionally connected subnetworks of neurons preferentially encoding harmonic over inharmonic sounds. These subnetworks are stable across days and exist prior to experimental harmonics exposure, suggesting their formation during development. Furthermore, A2 inactivation impairs performance in a discrimination task for coincident harmonics. Together, we propose A2 as a locus for multi-frequency integration, which may form the circuit basis for vocal processing.

[1] Department of Psychiatry, University of North Carolina at Chapel Hill, Chapel Hill, NC, USA. [2] Neuroscience Center, University of North Carolina at Chapel Hill, Chapel Hill, NC, USA. [3] Joint Department of Biomedical Engineering, University of North Carolina at Chapel Hill and North Carolina State University, Chapel Hill, NC, USA. [4] Carolina Institute for Developmental Disabilities, University of North Carolina at Chapel Hill, Chapel Hill, NC, USA.
✉email: hiroyuki_kato@med.unc.edu

O ur daily life critically depends on our ability to extract relevant information from a rich mixture of sensory inputs. Natural sounds, including language and animal vocalizations, contain a concurrent set of multiple frequencies, which are integrated by our brain and give rise to the perceived quality of sounds, such as timbre, pitch, and phonemes. How our brain integrates or segregates spectrally and temporally distributed input signals to reconstruct individual sound objects ("auditory scene analysis") remains a central question in systems neuroscience[1]. Psychophysical studies in humans have reported that one of the most prominent cues facilitating the integration of multifrequency sounds is the precise synchrony of their onset timings[1–3]. Frequencies that share common onset timings are more likely to be perceived as originating from the same source, whereas onset asynchrony as little as 30 ms results in the perception of separate sound objects. This strict temporal window for sound integration plays a crucial role in our recognition of phonemes in speech[4] and assures the segregation of auditory stimuli in a crowded sound environment[5]. Delineating the neural circuits underlying this synchrony-dependent feature binding is therefore fundamental for understanding how our brain encodes vocal communication.

In the peripheral auditory system, frequency components within sounds are decomposed into tonotopically organized narrow frequency channels. These channels ascend the auditory pathway in a parallel manner to reach the primary auditory cortex (A1), largely maintaining their tonotopic segregation. To form a coherent perception of the sound object, the central auditory system must integrate information across these frequency channels to reassemble the original spectral structure. In mammals, neurons that preferentially respond to multifrequency sounds have been observed in subcortical brain structures, including inferior colliculus[6,7] and thalamus[8]. Nevertheless, the auditory cortex is an especially attractive candidate for the locus of frequency integration, considering its role in processing complex sounds[9–11], as well as the extensive excitatory and inhibitory lateral connectivity across frequency channels[12–15]. Indeed, previous studies have found neurons in both primary and higher-order cortices that show preferential firing to combinations of frequencies over simple pure tones[16–20]. However, it remains unknown whether the activity of these neurons shares the same critical properties of perceptual binding in human psychophysics, such as synchrony dependence, and how they contribute to perceptual behaviors. To address this gap in knowledge and identify the primary locus of integration, we focused on harmonics, a key sound feature in animal vocalizations that consists of integer multiples of a fundamental frequency (F0). Taking advantage of two-photon calcium imaging, optogenetics, and electrophysiology in awake mice, we investigated the circuit mechanisms that enable perceptual binding of behaviorally relevant harmonic sounds.

## Results

### A2 neurons exhibit preferential responses to coincident harmonics, which deteriorate with shifts in component onset timing. To design harmonic stimuli that are representative of what mice experience in their natural environment, we first determined the range of F0s used in their harmonic vocalizations. Pain vocalizations, which predominantly contain harmonic structures[21,22], were recorded from three strains (C57BL/6J, BALB/c, and CBA) to capture their natural variability (Fig. 1a and Supplementary Fig. 1a). Harmonic template matching to recorded sounds showed that 80% of F0s fell between 1.8 and 4.1 kHz (median 3.2 kHz). Based on this observation, we designed sets of harmonic stimuli with F0s between 2 and 4 kHz, and also

included F0s outside this range to represent other harmonic sounds mice could encounter in their natural environment.

We next determined the cortical area that is most responsive to harmonic sounds, using macroscopic functional imaging of all auditory cortical areas. Individual auditory areas, including A1, A2, ventral auditory field (VAF), and anterior auditory field (AAF), were mapped by intrinsic signal imaging through the skull using pure tones (Fig. 1b). In contrast to the preferential activation of primary areas by pure tones, playback of a recorded harmonic vocalization (3.8 kHz F0)[23] reproducibly triggered strong responses in A2 (Fig. 1c–e). Measuring the ratio of vocalization to pure tone responses highlighted the inter-area difference, with the strongest contrast observed between A1 and A2 (A1: $0.99 \pm 0.04$; A2: $1.96 \pm 0.09$; $p < 0.0001$), and AAF being intermediate ($1.51 \pm 0.09$; A1 vs AAF: $p < 0.0001$; A2 vs AAF: $p < 0.0001$; see Supplementary Data 1 for additional statistics). Since our results were indistinguishable between naive males and females (Supplementary Fig. 1e; $p = 0.813$), they were combined for the remainder of the experiments. To examine if preferential activation of A2 is specific to this vocalization or generalized to harmonic structures, we next presented artificially generated harmonics (F0 = 2, 4, and 8 kHz, harmonic stacks up to 40 kHz). The strongest response was again observed in A2, confirming that both natural and artificial harmonics are preferentially represented in this higher-order cortex (Fig. 1f–h; harmonics/tone ratio: 2 kHz F0, A1: $1.09 \pm 0.06$; A2: $1.81 \pm 0.08$, $p < 0.0001$; 4 kHz F0, A1: $1.00 \pm 0.07$; A2: $1.94 \pm 0.11$, $p < 0.0001$; 8 kHz F0, A1: $0.84 \pm 0.04$; A2: $1.68 \pm 0.10$, $p < 0.0001$).

To identify the neural correlates of timing dependence in the perceptual binding of multifrequency sounds, we asked if A2 cellular responses to harmonics depend on the synchrony of component onsets. Two to three weeks following injection of GCaMP6s-expressing virus and glass window implantation, we performed targeted two-photon calcium imaging of layer 2/3 (L2/3) pyramidal cells in A1 and A2 of awake mice, guided by intrinsic signal imaging (Fig. 2a, b; see "Methods"). Timing dependence was determined by presenting artificial harmonics with onset shifts (Δonset: timing of lower compared to upper half, −45 to 45 ms, 15 ms steps) between the lower- and upper-halves of their components. Notably, we observed sharply tuned firing of A2 neurons to coincident (Δonset = 0 ms) harmonics, which deteriorated with Δonsets as little as 15 ms, mirroring perceptual binding in human psychophysics experiments (Fig. 2c–e). In contrast, A1 neurons overall showed Δonset-independent, or even shift-preferring, responses. To quantify timing dependence in individual neurons, we calculated the coincidence-preference index (CI; see "Methods"). Across all F0s, A2 neurons had a significantly higher CI than A1 (A1: $−0.09 \pm 0.02$; A2: $0.17 \pm 0.02$, $p < 0.0001$) (Fig. 2f). Taken together, these data suggest a role for A2 in integrating multifrequency sounds with synchronous onsets.

There are two potential explanations for the preferential representation of coincident harmonics in A2. First, coincident and shifted harmonics may activate the same neurons, but Δonsets simply affect the gain of response magnitudes uniformly across the population. Alternatively, these sounds may trigger different ensemble response patterns, potentially explaining their distinct perceptual quality. To test these two scenarios, we analyzed how ensemble representations of harmonics vary according to Δonset. We first visualized the data by projecting ensemble activity onto the first three principal components (Fig. 2g). Representations of harmonics in A1 neurons continuously changed across negative to positive Δonsets, as expected from the incremental change in the sound stimuli. In contrast, A2 harmonic representations showed abrupt transitions around Δonset = 0, which resulted in three discrete groups of data points—responses to negative Δonsets, positive Δonsets, and coincident harmonics. To quantify these

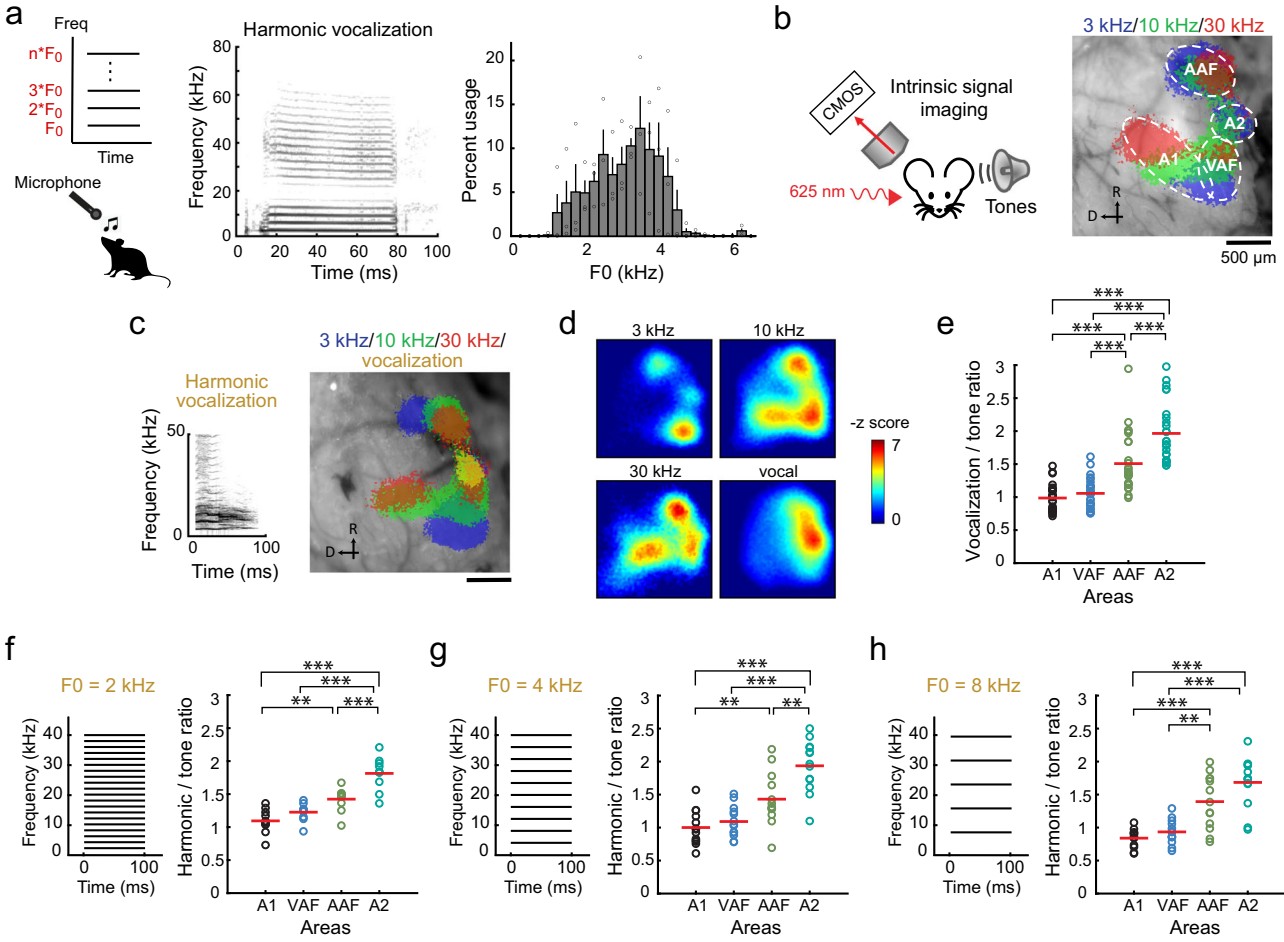

**Fig. 1 A2 preferentially responds to both natural and artificial harmonic sounds. a** Top left, schematic of a harmonic structure with its fundamental frequency (F0). Bottom left, vocalization recording schematic. Middle, spectrogram of a representative vocalization. Right, histogram showing the usage probability of F0 for harmonic vocalizations across C57BL/6J ($n = 5$ mice), BALB/c ($n = 4$ mice), and CBA ($n = 3$ mice). Results show mean ± SEM, overlaid with data points for individual strains. **b** Left, intrinsic signal imaging setup. Right, thresholded responses to pure tones superimposed on cortical surface imaged through the skull in a representative mouse. We observed similar tonotopy in all 62 mice across experiments. **c** Left, spectrogram of a harmonic vocalization[23] used for mapping. Right, thresholded intrinsic signals including harmonic vocalization for a representative mouse. $n = 26$ mice. **d** Heat maps showing z-scored response amplitudes in the same mouse as in (**c**). **e** Ratio of vocalization to pure tone response amplitudes in each auditory cortical area. $n = 26$ mice, ***$p < 0.001$ (one-way ANOVA followed by Tukey's HSD test). Red lines show mean. **f** Left, spectrogram of artificial 2 kHz F0 harmonics. Right, ratio of harmonics to pure tone response amplitudes in each auditory cortical area. $n = 10$ mice. **$p < 0.01$ (one-way ANOVA followed by Tukey's HSD test). **g** Data for 4 kHz F0 harmonics. $n = 13$ mice. **h** Data for 8 kHz F0 harmonics. $n = 13$ mice. See Supplementary Data 1 for additional statistics.

transitions in ensemble activity patterns, we calculated the pairwise Euclidean distance and correlation coefficient between data points in high-dimensional space (Fig. 2h–k). Both measures across all F0s showed that A2 representation of coincident harmonics was distinctly far away from those of temporally shifted harmonics. Therefore, A2 responses to coincident harmonics show not only large magnitude but also distinct ensemble patterns from those evoked by negative or positive Δonsets. A1 ensemble activity patterns also varied with Δonsets, but rather the patterns gradually changed across sound conditions. Thus, harmonics with a range of Δonsets activate distinct ensemble patterns in both A1 and A2, ruling out the gain-control hypothesis. However, only A2 shows a discrete activity pattern for coincident harmonics, suggesting a link between its activity and the perceptual binding of simultaneous sound components.

**Somatostatin-expressing neuron (SOM cell)-mediated inhibition limits the temporal window for harmonics integration.** What circuit mechanisms regulate the narrow temporal window

in which harmonic components are integrated in A2? To obtain insights into the temporal interplay between the leading and lagging components within harmonics, we compared the ensemble representations of temporally shifted harmonics with the lower- and upper-halves of their components (Fig. 3a). By visualizing the data in principal component space, we found that the ensemble representations of lower-leading harmonics (Δonset < 0) were close to that of the lower component itself, whereas upper-leading harmonics (Δonset > 0) clustered together with the upper component. This leading sound dominance of harmonic representations was evident when we quantified the pairwise Euclidean distance and correlation coefficient between data points (Fig. 3b, c). Across F0s, both measures showed that A2 representation of coincident harmonics was distinctly far away from those of component sounds, and there was an asymmetry between the contribution of leading and lagging components in both A1 and A2 (leading vs lagging, Euclidean distance, A1: $p = 0.0073$; A2: $p = 0.0001$; correlation coefficient, A1: $p < 0.0001$; A2: $p < 0.0001$). This diminished influence of lagging components suggests that leading sounds actively suppress the

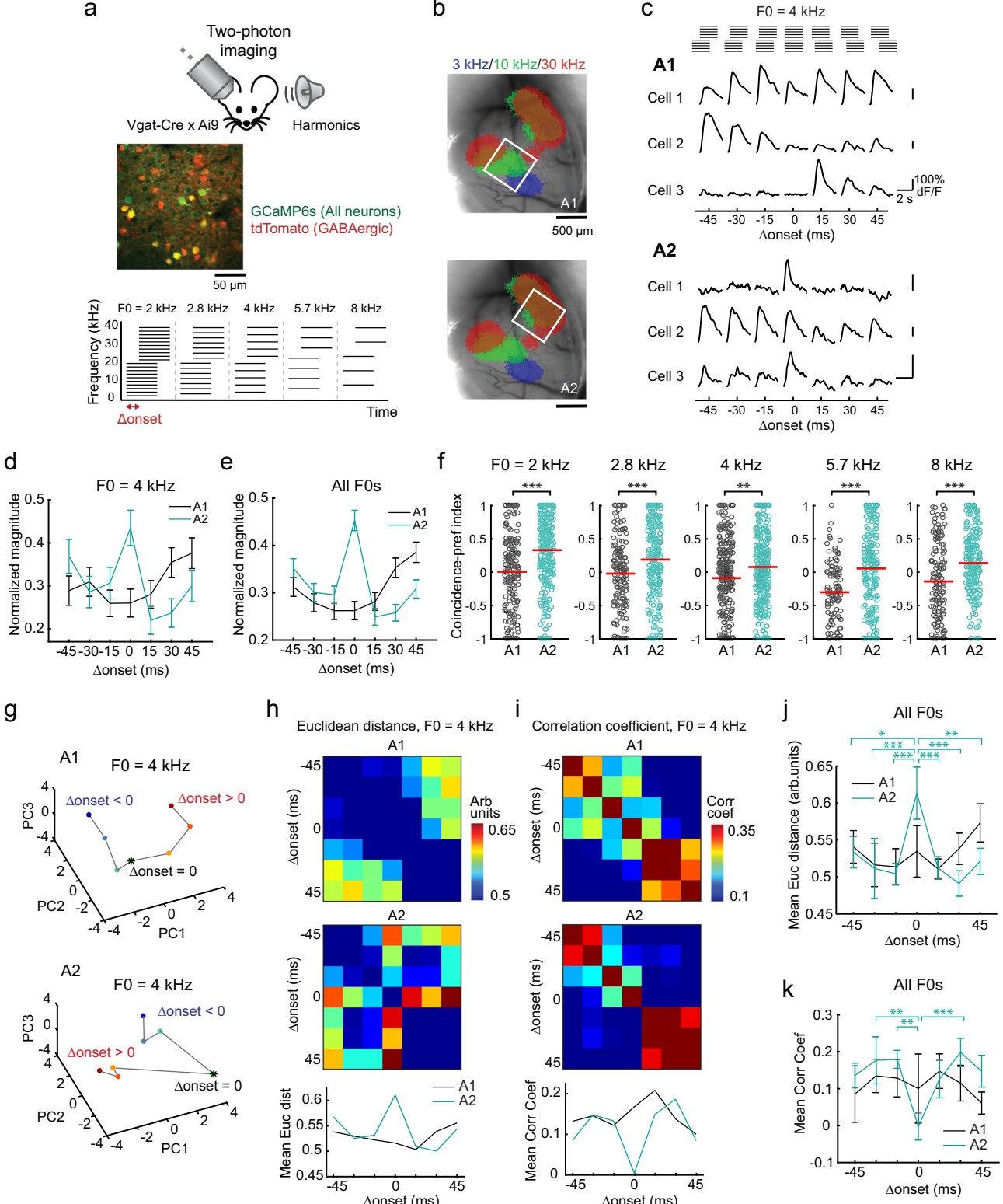

responses to lagging sounds, thus preventing the integration of temporally shifted harmonics.

To further examine the hypothesis that forward suppression limits the temporal window for harmonic integration in A2, we next focused on the role of cortical inhibition. Specifically, since recent studies found that SOM cells mediate delayed suppression in the auditory cortex[12,24], we tested whether inactivation of

different interneuron subtypes affects coincidence preference in pyramidal cells. We performed multiunit recordings in A2 superficial layers of head-fixed awake mice and photoinactivated SOM or parvalbumin-expressing neurons (PV cells) using virally introduced halorhodopsin (eNpHR3.0). Remarkably, SOM cell photoinactivation enhanced responses of regular-spiking units to shifted harmonics without largely altering their responses

**Fig. 2 Coincident and shifted harmonics recruit distinct neuronal ensemble patterns in A2. a** Top, representative in vivo two-photon image of L2/3 neurons in A2. Bottom, schematic of artificial harmonics with Δonsets. $n = 10$ mice. **b** Intrinsic signal image superimposed on cortical vasculature imaged through a glass window. White squares represent the two-photon imaging fields of view in A1 (top, $n = 11$) and A2 (bottom, $n = 10$ mice). **c** Responses of representative L2/3 pyramidal cells in A1 and A2 to harmonics with various Δonsets. Traces are average responses (five trials). **d** Summary data comparing normalized responses to coincident and shifted 4 kHz F0 harmonics in A1 and A2. A1: $n = 11$ mice, 144 cells; A2: $n = 10$ mice, 119 cells. **e** Summary data comparing normalized response to coincident and shifted harmonics across all F0s. A1: $n = 401$ cell–F0 pairs; A2: 431 cell–F0 pairs. **f** Coincidence-preference index (CI) for each F0 in A1 and A2. A1: $n = 182, 157, 299, 104, 145$ cell–Δonset pairs; A2: $n = 278, 215, 303, 181,$ and 188. ***$p < 0.001$, **$p < 0.01$, *$p < 0.05$ ($p = 1.57 \times 10^{-10}$, $8.21 \times 10^{-4}$, $4.22 \times 10^{-4}$, $1.35 \times 10^{-7}$, and $4.31 \times 10^{-5}$ for each F0, two-sided Wilcoxon's rank-sum test). Red lines show mean. **g** Principal component analysis for L2/3 pyramidal cell responses to coincident and shifted harmonics (4 kHz F0) in A1 and A2 ($n = 144$ and 119 responsive cells). Color gradient from blue to red represents data points with various Δonsets. Asterisks, coincident harmonics. **h** Matrices depicting the pairwise Euclidean distance between ensemble activity patterns evoked by 4 kHz F0 harmonics with various Δonsets in A1 (top) and A2 (middle). Bottom: mean Euclidean distance between each Δonset and all others. **i** Correlation coefficient data for 4 kHz F0 harmonics. **j, k** Summary data showing (**j**) the mean Euclidean distance ($n = 5$ F0s; Δonset, $p = 3.24 \times 10^{-5}$; A1 vs A2, $p = 0.286$; interaction, $p = 0.0007$) and (**k**) the mean correlation coefficient in A1 and A2 ($n = 5$ F0s; Δonset, $p = 0.0029$; A1 vs A2, $p = 0.0461$; interaction, $p = 0.0347$, two-way ANOVA followed by Tukey's HSD test). Results show mean ± SEM in (**d, e, j,** and **k**). See Supplementary Data 1 for additional statistics.

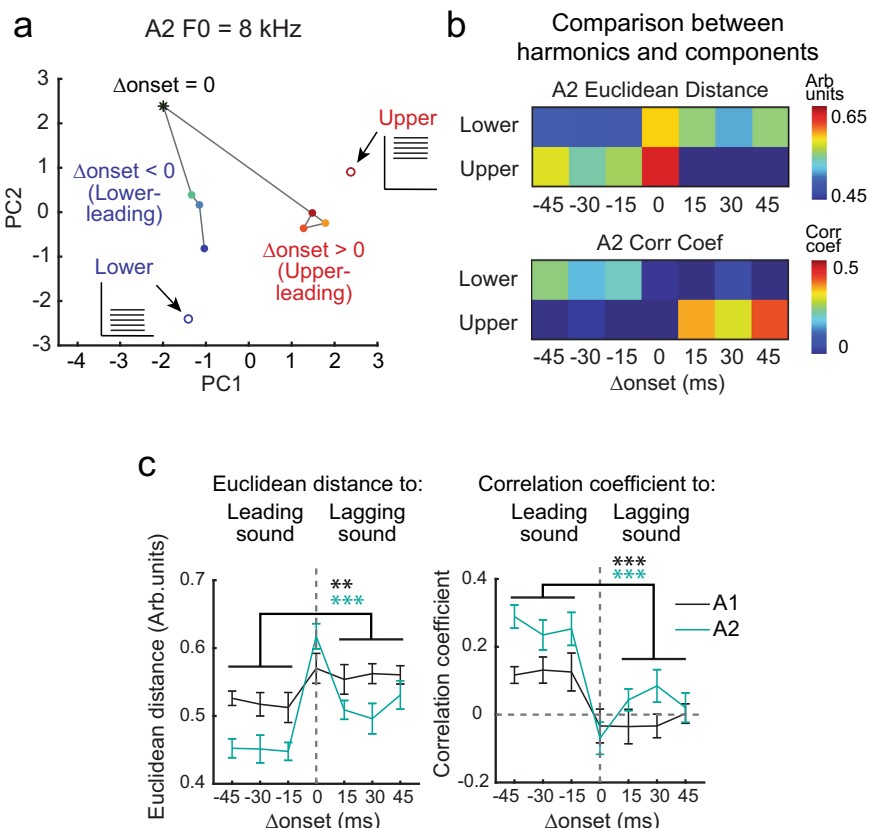

**Fig. 3 Neuronal representations of shifted harmonics are dominated by leading group of tones. a** Principal component analysis for A2 neuronal responses to coincident and shifted harmonics (8 kHz F0) as well as their lower and upper components ($n = 74$ responsive cells). Color gradient of closed circles represents data points with various Δonsets. Asterisk, coincident harmonics. Blue and red open circles, lower and upper components. **b** Top, matrix depicting the pairwise Euclidean distance between ensemble activity patterns evoked by temporally shifted 8 kHz F0 harmonics and their components. Bottom, matrix depicting the pairwise correlation coefficient. **c** Left, summary plot showing the Euclidean distance between ensemble activity patterns evoked by temporally shifted harmonics and their leading and lagging components. Right, summary plot showing the correlation coefficient. Data are pooled across all F0–component pairs ($n = 10$) and shown separately for A1 and A2. Results are mean ± SEM. ***$p < 0.001$, **$p < 0.01$ (leading vs lagging, Euclidean distance, A1: $p = 0.0073$; A2: $1.33 \times 10^{-4}$; correlation coefficient, A1: $p = 5.27 \times 10^{-5}$; A2: $1.11 \times 10^{-7}$; two-sided Wilcoxon's rank-sum test).

to coincident harmonics (Fig. 4a–e). As a result, CI was significantly reduced in Light Emitting Diode (LED) trials compared to interleaved control trials (Fig. 4e; $p = 0.0422$). In contrast, photoinactivation of PV cells increased the overall activity of A2 neurons to all sounds (Fig. 4f–j) and did not alter CI (Fig. 4j; $p = 0.457$). The difference between SOM and PV photoinactivation effects cannot be attributed to different levels of optogenetic manipulation, as quantified by the increase in

spontaneous firing rate (Supplementary Fig. 2a, b). Photoinactivation of either SOM or PV cells had no effect on the coincidence preference of A1 neurons, as they were indifferent to Δonsets in the control conditions (Supplementary Fig. 2c–f). Taken together, these results demonstrate that SOM cells limit the temporal window for multifrequency integration, likely through forward suppression of non-coincident sounds, while PV cells regulate the gain of responses. These data also rule out the possibility that

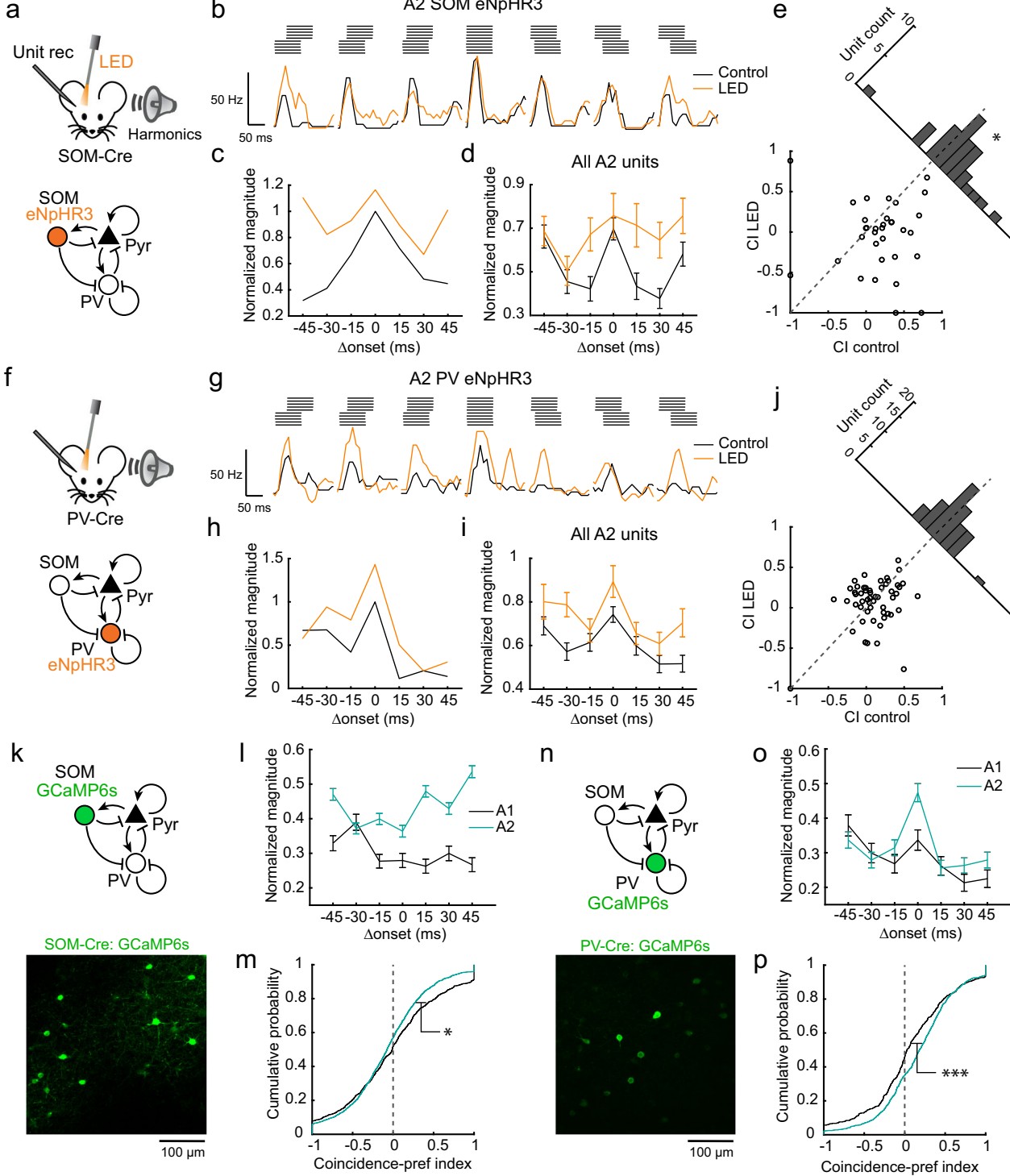

**Fig. 4 SOM cell-mediated inhibition determines the temporal window for harmonics integration. a** Schematics of optogenetic inactivation of SOM cells during multiunit recording. **b** Peristimulus time histogram (PSTH) of a representative regular-spiking unit to 4 kHz F0 harmonics with various Δonsets during control and SOM cell inactivation trials. **c** Response amplitudes of the unit in (**b**) normalized to the maximum response amplitude in the control condition. **d** Normalized response amplitudes averaged across all units in A2 ($n = 6$ mice, 33 units). **e** Scatter plot showing CI during control and SOM cell inactivation trials. The oblique histogram illustrates the changes in CI with LED. *$p = 0.0422$ (two-sided paired $t$ test). **f–j** Responses to temporally shifted harmonics with and without PV cell photoinactivation ($n = 7$ mice, 54 units; $p = 0.457$, two-sided paired $t$ test). **k** Representative in vivo two-photon image of GCaMP6s-expressing L2/3 SOM cells. $n = 6$ mice. **l** Summary data comparing normalized response of SOM cells to coincident and shifted harmonics across all F0s. A1: $n = 6$ mice, 360 cell–F0 pairs; A2: $n = 5$ mice, 538 cell–F0 pairs. **m** CI of SOM cells pooled across all F0s shown as cumulative probability plots. A1: $n = 848$; A2: $n = 1598$ cell–F0–Δonset combinations. $p = 0.0110$ (two-sided Wilcoxon's rank-sum test). **n** Representative in vivo two-photon image of GCaMP6s-expressing L2/3 PV cells. $n = 8$ mice. **o** Summary data comparing normalized response of PV cells to coincident and shifted harmonics across all F0s. A1: $n = 8$ mice, 223 cell–F0 pairs; A2: $n = 6$ mice, 310 cell–F0 pairs. **p** CI of PV cells. A1: $n = 476$; A2: $n = 844$ cell–F0–Δonset combinations. ***$p = 1.02 \times 10^{-4}$ (two-sided Wilcoxon's rank-sum test). Results show mean ± SEM in (**d**, **i**, **l**, and **o**).

coincidence preference in A2 is simply inherited from upstream subcortical systems and demonstrate active shaping of cortical temporal integration by inhibition.

To further characterize how inhibitory neuron subtypes contribute to the processing of coincident and shifted harmonics, we performed two-photon calcium imaging of GCaMP6s-expressing SOM and PV cells (Fig. 4k, n). Similar to pyramidal cells, PV cells exhibited strong coincidence preference in A2, but less in A1 (Fig. 4o, p; A1: $0.066 \pm 0.003$; A2: $0.171 \pm 0.006$; A1 vs A2: $p = 0.0001$), suggesting their co-modulation with surrounding pyramidal cells. In contrast, SOM cells overall exhibited $\Delta$onset-independent, or even shift-preferring, responses in both areas (Fig. 4l), quantified by a small but negative CI (Fig. 4m; A1: $-0.0062 \pm 0.0002$; A2: $-0.073 \pm 0.002$; A1 vs A2: $p = 0.003$). This lack of forward suppression in SOM cells is consistent with the observation that these cells do not inhibit each other[25]. Therefore, SOM cell activity regardless of $\Delta$onset ensures the robust suppression of lagging sounds and limits the temporal window for the integration of non-coincident sounds in pyramidal cells.

**Coincident harmonics-preferring neurons form functional subnetworks in A2.** Although SOM cell-mediated forward suppression shapes the narrow temporal integration window in A2 neurons, inhibitory mechanisms alone are insufficient to account for the differential responses to coincident harmonics between areas (Fig. 1c–h). Therefore, we sought additional circuit mechanisms by further characterizing A2 pyramidal cell responses to harmonics. Motivated by the discrete representations of coincident and shifted harmonics observed in A2 (Fig. 2g–k), we used unsupervised clustering to group neurons depending on their responses to harmonics with various $\Delta$onsets. Surprisingly, clustering of all responsive cell–F0 pairs combining A1 and A2 revealed a group of neurons exclusively responsive to coincident harmonics. Overall, we found three clusters representing cells preferring negative $\Delta$onsets, positive $\Delta$onsets, and coincident harmonics (Fig. 5a and Supplementary Fig. 3), and the fraction of coincidence-preferring cells was 2.2-fold larger in A2 compared to A1 (Fig. 5b; all F0 combined: A1: 11.5%; A2: 25.8%; $p < 0.0001$). This overrepresentation of coincidence-preferring cells in A2 likely explains the abrupt transitions of ensemble representations around $\Delta$onset $= 0$ in this area, while those in A1 transitioned more continuously (Fig. 2).

Having identified coincident harmonics-preferring neurons, we asked if they are spatially clustered within A2. Coincident harmonics-preferring neurons for five F0s were spatially inter-mingled with each other and represented partially overlapping populations in A2 (Fig. 5c, d). Coincidence-preferring cell pairs within the same F0 were significantly closer together than random cell pairs (Fig. 5e). Likewise, cell pairs across F0s were also spatially clustered together, demonstrating intermingled and restricted localization of coincidence-preferring cells of different F0s (same F0: $p < 0.0001$; across F0: $p < 0.001$; same vs across: $p = 0.0486$). This spatial clustering was not observed in A1, further suggesting distinct circuit organization between the two areas (Supplementary Fig. 4a, b). Although harmonics-responding neurons tended to have broader pure tone tuning than harmonics-nonresponding neurons, we observed no difference between the tuning broadness of coincidence- and shift-preferring neurons (Supplementary Fig. 5a–c), suggesting that coincidence preference is not simply conferred by the neurons' tuning properties. We also did not observe a relationship between preferred F0 and characteristic frequency (CF) of coincident harmonics-preferring neurons, suggesting that these neurons do not encode pitch[26] (Supplementary Fig. 5d).

Integration of complex sound features may be achieved by specialized excitatory connectivity within cortical circuits. To test this, we asked whether coincident harmonics-preferring neurons in A2 form functional subnetworks by measuring the pairwise noise correlation between simultaneously imaged neurons during the presentations of harmonic stimuli. Notably, coincident harmonics-preferring cell pairs within the same-F0 cluster displayed a significantly higher noise correlation than pairs across different F0s or across random cell pairs whose spatial distances were matched to those of same-F0 pairs (Fig. 5f, same F0 vs random: $p < 0.0001$; across F0s vs random: $p = 0.0009$; same vs across: $p = 0.0287$). Differences in noise levels or signal-to-noise ratios (SNRs) across cells did not account for the higher noise correlation observed within the same-F0 cluster (Supplementary Fig. 4c, d). These data suggest the existence of F0-specific functional subnetworks of coincidence-preferring neurons in A2, which is not simply a consequence of their spatial proximity or activity level.

Did the observed A2 subnetworks emerge in response to experimental exposure to harmonic sounds or did our experiment tap into preexisting functional circuits? To examine this, we focused on the clusters of neurons we identified during the harmonics experiments but measured noise correlation using pure tone responses collected from the same neurons on the previous day. Strikingly, in A2, pairs of neurons that would eventually be clustered together as coincidence-preferring to individual F0s already had higher noise correlation compared to across F0s and distance-matched pairs (Fig. 5g, same F0 vs random: $p = 0.0001$; across F0s vs random: $p = 0.346$; same vs across, $p = 0.0735$). When noise correlation in individual same-F0 cell pairs was compared between harmonics and pure tone experiments, these values showed a significant positive correlation with each other, supporting an overlapping set of cells in the subnetwork across days (Supplementary Fig. 4e). Therefore, coincident harmonics-preferring neurons in A2 form F0-specific functional subnetworks that are stable across days and exist even before our harmonics experiments, suggesting that A2 is pre-equipped with circuitry fine-tuned to process ethologically relevant multifrequency sounds.

**A2 subnetworks are preferentially recruited by sounds with harmonic regularity.** Harmonic regularity within multifrequency sounds is itself a cue for perceptual binding[27,28]. Therefore, we next asked if the functional subnetworks we identified in A2 are specific to harmonic sounds or generalized to inharmonic sounds. To answer this, we introduced spectral jitters to ten-tone harmonics (4 kHz F0, Fig. 6a) and examined how they affect cortical harmonic representations. We first conducted intrinsic signal imaging of all auditory cortical areas using both harmonic and jittered sounds and found that both sounds preferentially activated A2 (Fig. 6a, Supplementary Fig. 6a; multifreq/tone ratio, harmonic: $1.94 \pm 0.13$; jittered: $1.79 \pm 0.29$, $p = 0.278$). To further investigate whether coincidence-preference exists for jittered sounds in individual A2 neurons, we again turned to two-photon calcium imaging and compared $\Delta$onset-dependent responses between harmonic and jittered sounds (Fig. 6b). Although preference for harmonic vs jittered sounds varied in individual neurons, coincidence preference in A2 was observed for both sounds as a population (Fig. 6b–d; CI, harmonic: A1: $-0.04 \pm 0.03$, A2: $0.22 \pm 0.03$, $p < 0.0001$; jittered: A1: $-0.10 \pm 0.03$, A2: $0.19 \pm 0.03$, $p < 0.0001$). Thus, coincidence-dependent integration in A2 was robust regardless of spectral structure, which is crucial for the integration of natural sounds that are not purely harmonic. Unsupervised clustering revealed spatial localization of coincidence-preferring neurons in A2 for both harmonic and inharmonic sounds (Fig. 6e–i and Supplementary Figs. 3b and 6b,

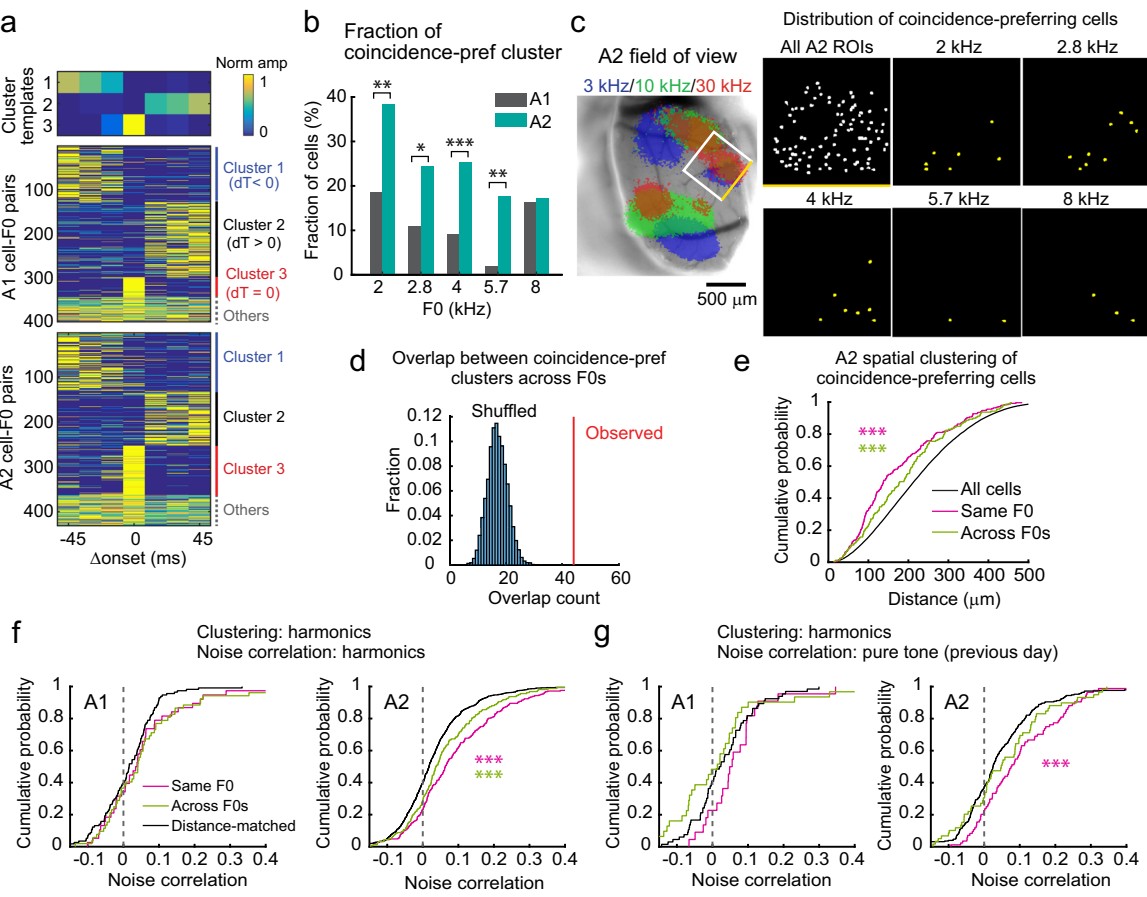

**Fig. 5 Coincident harmonics-preferring A2 neurons form F0-specific functional subnetworks. a** Clustering of A1 and A2 neural responses into negative shift-, positive shift-, and coincidence-preferring groups. A1: n = 401 cell–F0 pairs in 11 mice; A2: n = 431 cell–F0 pairs in ten mice. Top, cluster templates determined by non-negative matrix factorization (NMF). **b** Fraction of cells that fell into the coincidence-preferring cluster for each F0 in A1 and A2. ***p < 0.001, **p < 0.01, *p < 0.05 (p = 0.0055, 0.0326, 0.0004, 0.0066, 0.9060 for each F0, two-sided $\chi^2$ test). **c** Spatial distribution of coincidence-preferring cells for each F0 in A2 of a representative mouse. Left, intrinsic signal image. White square represents two-photon imaging field of view. One of the edges is shown in yellow for alignment. Right, maps showing the location of imaged pyramidal cells that fell into the coincidence-preferring cluster for each F0. Similar results were observed in ten mice for A2 imaging. **d** Count of overlap between coincidence-preferring cells across F0s for A2. Red line, observed overlap count (i.e., cells counted in the coincidence-preferring cluster for multiple F0s); histogram, distribution of overlap count for shuffled data (10,000 repetitions, p < 0.0001, permutation test). **e** Cumulative probability plot of spatial distance between all cells, between coincidence-preferring cells with the same F0, and between coincidence-preferring cells across F0s in A2 (n = 28,439, 248, and 159 for All cells, same F0s, and across F0s). ***p < 0.001 (two-sided Wilcoxon's rank-sum test). **f** Cumulative probability plots of noise correlation between coincidence-preferring cell pairs during presentation of harmonics (A1: n = 38, 52, 114; A2: n = 248, 159, 744 for same F0, across F0s, and distance-matched). **g** Cumulative probability plots of noise correlation between coincidence-preferring cell pairs during presentation of pure tones (A1: n = 22, 31, 66; A2: n = 84, 59, 252 for same F0, across F0s, and distance-matched). See Supplementary Data 1 for additional statistics.

c). Surprisingly, despite the similar spatial clustering of coincidence-preferring neurons for harmonic and jittered sounds, the noise correlation for cell pairs in the jittered-sound cluster was significantly lower than that of the harmonic-sound cluster (Fig. 6j and Supplementary Fig. 6, p = 0.0046), indicating that coincidence-preferring neurons for jittered sounds form less tightly coupled subnetworks. We note that the noise correlation between the neurons responding to jittered sounds was significantly higher than between distance-matched random cell pairs (p = 0.002) or noise level-matched cell pairs (Supplementary Fig. 6f, g). The above-chance overlap between coincidence-preferring cells for harmonic and jittered sounds (Fig. 6h) likely accounts for this observation and may suggest that jittered harmonics recruit harmonics-encoding subnetworks through a pattern completion mechanism in A2 circuits.

**Responses to harmonics exhibit nonlinear summation of their component pure tones.** We next asked what computation takes place in individual neurons during harmonic integration in A2. To this end, we examined the linearity of integration by measuring responses to both three-tone harmonics (Fig. 7a) as well as their component pure tones. Consistent with our previous results, we observed coincidence-preferring responses in A2, but not in A1 (Fig. 7b). For each neuron, we calculated the linearity index (LI; see "Methods") for each Δonset and observed overall sublinear summation in both areas. However, an increase in LI was observed for coincident harmonics only in A2, indicating a timing-dependent change in the mode of summation (Fig. 7c, d; LI, A1 coincident: −0.30 ± 0.02, A1 shifted: −0.30 ± 0.02, p = 0.920; A2 coincident: −0.14 ± 0.03, A2 shifted: −0.30 ± 0.02, p < 0.0001).

We also performed in vivo whole-cell recordings in awake mice to measure synaptic inputs onto individual neurons, which contribute to coincidence-dependent firing (Fig. 7e). We targeted recording pipettes to L2/3 of A1 or A2, guided by intrinsic signal imaging, and performed voltage-clamp recording to measure excitatory postsynaptic currents (EPSCs).

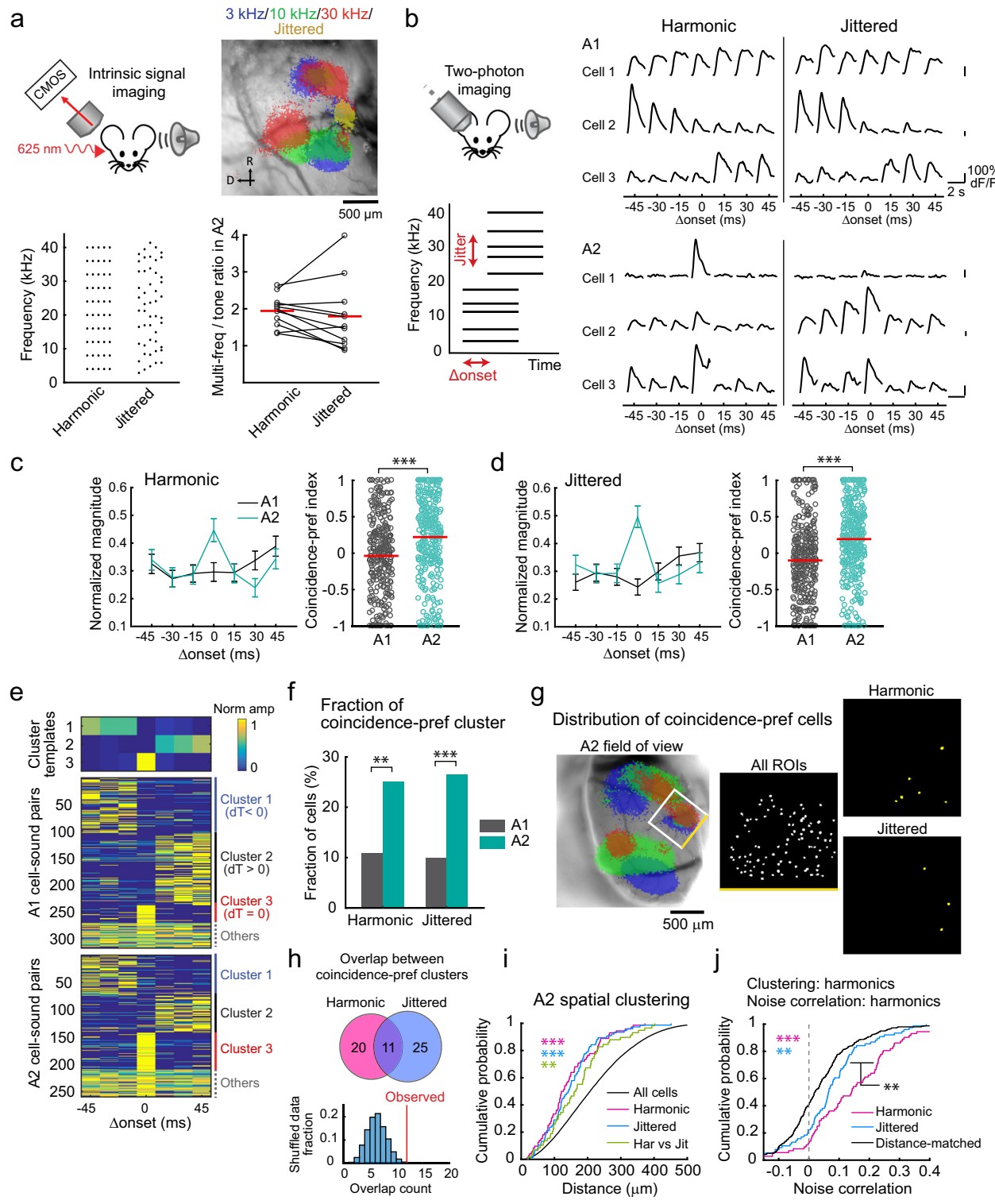

Notably, quantification of excitatory charge triggered by three-tone harmonics showed that, similar to neuronal output, synaptic inputs onto A2 neurons already display coincidence preference (Fig. 7f). A2 neurons had significantly higher LI of EPSCs for coincident compared to shifted harmonics (Fig. 7g, h; A1: coincident = $-0.40 \pm 0.05$, shifted = $-0.40 \pm 0.04$, $p = 0.979$; A2: coincident = $-0.13 \pm 0.08$, shifted = $-0.28 \pm 0.06$, $p = 0.0072$), and supralinear integration was observed in 4 out of

13 neurons in A2, in contrast to 0 out of 11 in A1. Similar coincidence preference in inhibitory postsynaptic currents was observed in A2, but not in A1, consistent with the inhibition-stabilized network operation of the auditory cortex[12,29,30] (Supplementary Fig. 7). The fact that coincidence-preference and supralinear summation exists at synaptic inputs in A2, but not upstream in A1, argues against the simple explanation that the narrow temporal window for integration in A2 neurons is

**Fig. 6 A2 subnetworks are preferentially recruited by sounds with harmonic regularity. a** Top left, intrinsic signal imaging setup. Bottom left, five example frequencies used for harmonic and jittered-sound presentations. Top right, thresholded responses to pure tones as well as jittered harmonics superimposed on cortical surface imaged through the skull in a representative mouse. Bottom right, ratio of harmonic or jittered sounds to pure tone response amplitudes in A2. $n = 11$ mice, $p = 0.278$ (two-sided Wilcoxon's signed-rank test). **b** Left, schematic for two-photon calcium imaging setup to test coincidence preference for harmonic and jittered sounds. Right, responses of representative L2/3 pyramidal cells in A1 and A2 for harmonic and jittered sounds with various Δonsets. Traces are average responses (five trials). **c** Summary data comparing normalized responses to coincident and shifted 4 kHz F0 harmonics in A1 and A2. A1: $n = 147$ cells in 11 mice, A2: $n = 124$ cells in ten mice. Right, CI for 4 kHz F0 harmonics. ***$p = 8.83 \times 10^{-9}$ (two-sided Wilcoxon's rank-sum test). **d** Summary data comparing normalized responses to jittered sounds. A1: $n = 171$; A2: $n = 136$ cells. Right, CI for jittered sound. ***$p = 4.06 \times 10^{-13}$ (two-sided Wilcoxon's rank-sum test). Results show mean ± SEM in (**c**, **d**). **e** NMF clustering of A1 and A2 neural responses into negative shift-, positive shift-, and coincidence-preferring groups. A1: $n = 318$; A2: $n = 260$ cell–sound pairs. Data include responses to both harmonic and nonharmonic sounds. **f** Fraction of cells that fell into the coincidence-preferring cluster for harmonic and jittered sounds. **$p = 0.0022$, ***$p = 1.53 \times 10^{-4}$ (two-sided $\chi^2$ test). **g** Spatial distribution of coincidence-preferring cells for harmonic and jittered sounds in A2 of a representative mouse. Left, intrinsic signal image. Right, maps showing the location of imaged pyramidal cells that fell into the coincidence-preferring cluster for harmonic and jittered sounds. Similar results were observed in ten mice for A2 imaging. **h** Top, Venn diagram showing the overlap between coincidence-preferring cells for harmonic and jittered sounds. Bottom, observed overlap of coincidence-preferring clusters for harmonic and jittered sounds (red line) compared to shuffled data (histogram, 10,000 repetitions, $p = 0.0026$, permutation test). **i** Cumulative probability plot of spatial distance between coincidence-preferring cells in A2 ($n = 27,130, 72, 76, 58$ for All cells, Harmonic, Jittered, and Har vs Jit). ***$p < 0.001$, **$p < 0.01$ (two-sided Wilcoxon's rank-sum test). **j** Cumulative probability plot of noise correlation between coincidence-preferring cells in A2 ($n = 72, 76, 399$ for Harmonic, Jittered, and Distance-matched). See Supplementary Data 1 for additional statistics.

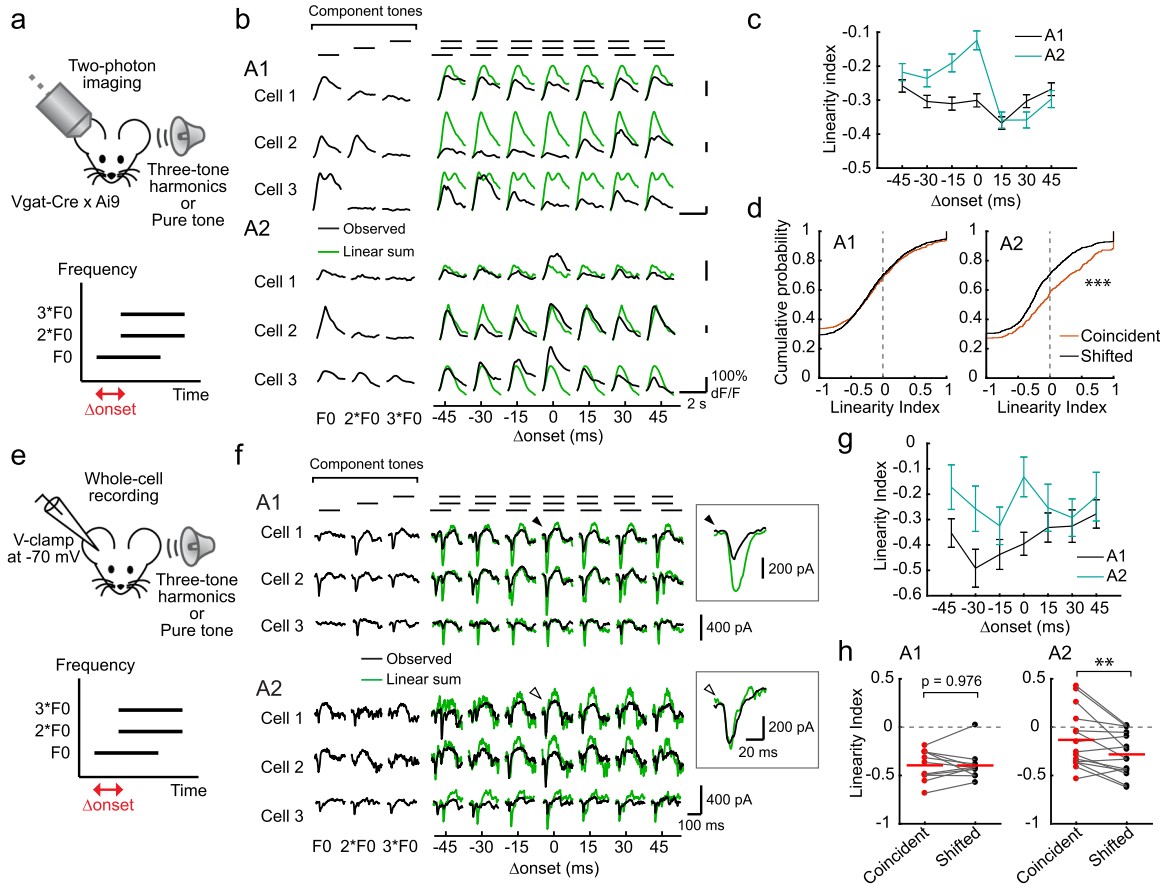

**Fig. 7 Responses to harmonic sounds exhibit nonlinear summation of component pure tones. a** Top, in vivo two-photon calcium imaging of L2/3 pyramidal cells in awake, head-fixed mice. Bottom, schematic for three-tone harmonic stimulus with a shift in onset timings. **b** Responses of representative L2/3 pyramidal cells to three-tone harmonics with various Δonsets as well as component pure tones in A1 and A2. Traces are average responses (five trials). Green traces show the linear sum of the responses to three component tones. **c** Linearity index (LI) calculated for each Δonset in A1 and A2. A1: $n = 11$ mice, 1094 cell–F0 pairs; A2: $n = 8$ mice, 640 cell–F0 pairs. **d** Cumulative distribution plots showing LI for coincident and shifted (±15 and 30 ms) harmonics, respectively. A1: $n = 722$ and 1404 cell–F0 pairs for coincident and shifted sounds, $p = 0.920$; A2: $n = 422$ and 734 cell–F0 pairs, ***$p = 1.71 \times 10^{-5}$ (two-sided Wilcoxon's rank-sum test). **e** Top, in vivo whole-cell recording of L2/3 pyramidal cells in awake, head-fixed mice. Bottom, schematic for three-tone harmonics stimulus with a shift in onset timings. **f** EPSCs of representative L2/3 pyramidal cells to three-tone harmonics with various Δonsets as well as component pure tones in A1 and A2. Traces are average responses (3–9 trials). Insets show magnified views of traces indicated by filled (A1) or open (A2) arrowheads. **g** LI calculated for each Δonset in A1 and A2 (A1: $n = 11$; A2: $n = 15$ cells). **h** Summary plots showing the LI calculated for coincident and shifted harmonics (A1: $n = 11$ cells, $p = 0.979$; A2: $n = 15$, **$p = 0.0072$, two-sided paired $t$ test). Red lines show mean. Results show mean ± SEM for (**c**, **g**).

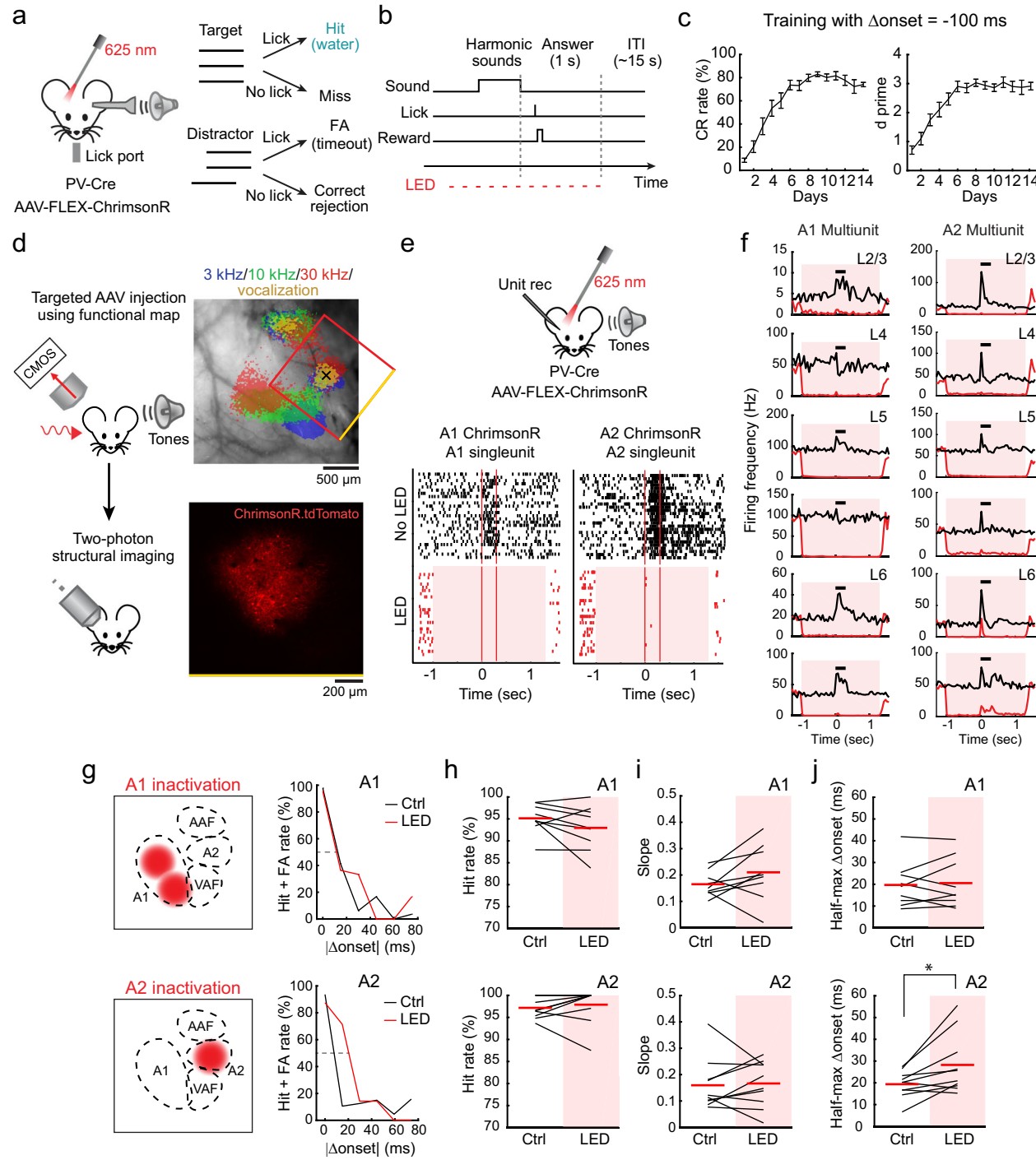

due to their intrinsic membrane properties and rather supports our finding of excitatory subnetworks within A2.

**A2 activity is causally related to performance in a harmonics discrimination task.** Finally, to determine whether A2 is required for the perception of integrated harmonics, we optogenetically inactivated A1 or A2 and examined the effects on perceptual behaviors (Fig. 8a–c). To achieve area-specific inactivation, ChrimsonR was expressed in PV cells using targeted viral injection in A1 or A2[31]. The spatial restriction of viral infection to individual areas was visualized in vivo by the coexpression of tdTomato (Fig. 8d). Extracellular recordings verified that LED illumination successfully inactivated one area without directly

suppressing the other area (Fig. 8e, f, Supplementary Fig. 8, and see "Discussion"; A1: spontaneous, −98 ± 1%, evoked, −89 ± 2%; A2: spontaneous, −96 ± 2%, evoked, −54 ± 2%).

For the behavior experiments, we used three-tone harmonics (4/8/12 kHz) that are confirmed to be perceived in an onset timing-dependent manner in mice[32]. We trained mice to lick for a water reward following coincident harmonics and to withhold licking for shifted harmonics (Δonset = ±100 ms) (Fig. 8a, b). Mice successfully learned to distinguish harmonics with negative Δonset in two weeks (Fig. 8c, correct rejection = 87 ± 2%; d prime = 3.2 ± 0.1). However, since they failed to perform the task for positive Δonset (Supplementary Fig. 9; day 14: d prime = 1.6 ± 0.1; correct rejection = 29 ± 4%), potentially due to masking of

**Fig. 8 A2 inactivation impairs performance of a harmonics discrimination task. a** Schematic for behavior setup. FA: false alarm. **b** Trial structure. Licking during a 1 s answer period immediately following the target offset triggers a water reward. On randomly interleaved trials (30%), auditory cortex is illuminated with LED pulses to inactivate A1 or A2. **c** Learning curves averaged across all tested mice (n = 19) during the training with an onset shift of −100 ms. Left, average correct rejection (CR) rate over days. Right, average d prime over days. Results show mean ± SEM. **d** Area-restricted expression of ChrimsonR-tdTomato in a representative mouse. Top, intrinsic signal image superimposed on cortical surface imaged through the skull. Black cross, AAV injection site. Red box, two-photon imaging field of view. Bottom, two-photon image of tdTomato fluorescence in A2 of the same mouse through a glass window. Similar restriction of expression was observed in n = 2 mice. **e** Top, Schematic of optogenetic inactivation of A1 or A2 during extracellular recording. Bottom, raster plots showing the responses of representative A1 and A2 singleunits to pure tones around its best frequency with (red) or without (black) LED. Red shading, timing of photostimulation. Red vertical lines, timing of sound presentation. Control and photostimulation trials were interleaved, but separated here for clarity. **f** PSTHs of multiunit spike counts across cortical layers during control and photostimulation trials for representative mice with A1 or A2 inactivation. Black bars indicate timing of sound presentation. **g** Left, schematics showing virus injection sites for A1 and A2 inactivation experiments, respectively. Right, discrimination psychometric curves against Δonset in representative mice with (black) and without (red) inactivation of A1 or A2. Dashed lines show 50% response rate. **h** Summary data showing hit rates for target sounds with and without inactivation of A1 (n = 9 mice, p = 0.129) or A2 (n = 10 mice, p = 0.469). Red lines show mean. **i** Summary data showing slopes of the fit lines for psychometric curves (see "Methods") with and without inactivation of A1 (p = 0.250) and A2 (p = 0.695). **j** Summary data showing half-max Δonsets with and without inactivation of A1 (p = 0.910) or A2 (*p = 0.0137). Two-sided Wilcoxon's signed-rank test for (**h–j**).

the delayed F0 by the generation of a combination tone from the upper components[33], we focused on negative Δonsets for the following experiments. On test day, stimuli with multiple Δonsets (−75 to 0 ms, 15 ms steps) were presented to determine the psychophysical threshold for discrimination. In 30% of randomly interleaved trials, the area expressing ChrimsonR was inactivated with LED illumination, and psychometric curves were compared between LED and control trials (Fig. 8g and Supplementary Fig. 10). While cortical inactivation did not affect the hit rate for the target sound (Fig. 8h, A1 control: 95.1 ± 1.1%, LED: 93.0 ± 1.7%, p = 0.129; A2 control: 97.2 ± 0.73%, LED: 97.9 ± 1.3%, p = 0.469), only A2 inactivation affected behavior via an increase in the false alarm rate. As a result, A2 inactivation shifted the psychometric curve to larger Δonsets, resulting in a 51.5 ± 0.2% broadening of the temporal window in which mice fail to discriminate coincident harmonics (Fig. 8i, j; slope, control: 0.16 ± 0.03, LED: 0.17 ± 0.03, p = 0.695; half-max Δonset, control: 19.4 ± 1.8 ms, LED: 28.2 ± 4.1 ms, p = 0.0137). Importantly, A1 inactivation did not alter perceptual discrimination (slope, control: 0.17 ± 0.02, LED: 0.21 ± 0.04, p = 0.250; half-max Δonset, control: 19.7 ± 3.2 ms; LED: 20.5 ± 3.6 ms, p = 0.910; 7.0 ± 0.1% broadening), indicating A2-specific contribution to this task. Taken together, these results indicate that A2 is causally related to perceptual discrimination of spectro-temporal structures, solidifying its role in coincidence-dependent integration of multifrequency sounds.

## Discussion

A central question in sensory processing is how our brain integrates sensory inputs to achieve coherent perception of the external world. Many attributes of this "sensory feature binding" are shared across sensory modalities; however, one aspect that makes auditory feature binding unique is its dependence on stimuli that rapidly fluctuate on the order of milliseconds. Although onset synchrony of frequency components is known to be a strong cue for this concurrent binding, neural circuits mediating this computation remain unknown. In this study, we identified mouse A2 as a locus of multifrequency integration, whose precise synchrony-dependence mirrors perceptual binding. Integration of coincident multifrequency sounds, a simple and prevalent sound feature, could be fundamental for the perception of more complicated sounds, such as language.

**Integration of coincident multifrequency sounds in A2.** A2 in mice possesses properties of a higher-order cortex, including long latencies, broad frequency tuning, and less-ordered tonotopic

arrangement[34–37] (but see refs. [38,39]). These data led to the view that A2 corresponds to the "belt region," which extracts complex sounds such as vocalizations in primates[40]. However, there have been few studies focusing on A2 function, likely due to the challenge in reliably targeting A2 stereotaxically. In this study, combining macroscopic functional mapping and targeted cellular-level imaging, we identified a population of coincident harmonics-preferring neurons that spatially cluster in a restricted area within A2. A2 ensemble representations of multifrequency sounds depend on the precise synchrony of their onset timings, which was in clear contrast to A1, where only a minor fraction of neurons showed synchrony dependence. Furthermore, A2, but not A1, inactivation impaired performance in a harmonics discrimination task, supporting its role in the multifrequency integration underlying perception. There is no doubt that spectral integration begins already at A1[16,26,41] or even in subcortical systems[6–8]. We indeed found a small fraction of coincidence-preferring neurons in A1, consistent with the gradient-like hierarchy across primary and higher sensory areas[42]. Nevertheless, our results demonstrate areal specializations and indicate a preferential involvement of A2 in perceptual binding and encoding of spectral components with synchronous onsets.

Noise correlation analyses indicated harmonics-preferring subnetworks within A2, which potentially contribute to perception through computations such as gamma oscillations and pattern completion (e.g., our ability to fill in missing fundamentals). Although an indirect measure, high noise correlation between neurons suggests that they could receive shared presynaptic inputs, form mutual connections[43], and/or indirectly interact through multisynaptic pathways. Interestingly, whether direct or indirect, these connectivity schemes could emerge from the "neurons that fire together wire together" mechanism[44] during development, as natural sounds often contain harmonics. The weaker recruitment of subnetworks for inharmonic sounds is consistent with this idea, since the specific sets of jittered frequencies used in our experiments are likely to be a first-time experience for animals. However, even though there are less prominent subnetworks for the novel sets of inharmonic frequencies, coincidence-dependent encoding was still robust in A2. This is of obvious importance for animal survival, since natural sounds are rarely strictly harmonic, and animals need the ability to bind/segregate sounds based on synchrony even in the face of novel frequency combinations. It would be of interest to examine whether the formation of subnetworks in A2 is influenced by various experiences, such as repeated exposure to specific sets of inharmonic frequencies or maternal exposures to pups[10,45].

**Segregation of non-coincident multifrequency sounds in A2.**
Our imaging results suggest the existence of specialized connectivity in A2 that allows more supralinear summation of components, and our whole-cell recording data support this excitatory subnetwork idea. However, these excitatory mechanisms alone are not sufficient to explain the narrow temporal window for the integration of multifrequency sounds. Using selective optogenetic inactivation of inhibitory neuron subtypes, we demonstrate that forward suppression through cortical SOM cells, but not PV cells, are required for the sharp tuning of A2 neurons to coincident sounds. This, for the first time, demonstrates how specific inhibitory neuron subtypes allow for extraction of complex sensory stimuli in higher-order sensory cortex. SOM cells' unique $\Delta$onset-independent activity is likely a result of their broad input connectivity[12] as well as their lack of inhibition onto themselves[25]. Our results are consistent not only with SOM cell-specific contributions to delayed suppression[12,24] but also with the "temporal coherence theory," which hypothesized a role of delayed mutual inhibition between component frequencies in the segregation of sound streams[46]. Although the temporal coherence theory was proposed for sound stream segregation, which evolves over larger time scales (seconds)[1,47,48], its mechanisms could partially share properties with those of the concurrent (single-presentation) integration/segregation investigated in this study. It will be of great interest to further investigate how A2 inhibitory circuits contribute to segregation of sounds over various time scales, which enables auditory scene analysis in our daily life.

**The role of A2 in perceptual behaviors.** We used area-restricted optogenetic inactivation and found a role of A2 in the performance of a harmonics discrimination task. A2 inactivation caused a 52% broadening of the range of $\Delta$onsets in which mice failed to distinguish from synchronous harmonics, suggesting A2's role in determining the temporal window for perceptual integration. Although A2 inactivation in mice did not totally abolish their discrimination ability, this is likely an underestimation: first, to ensure area specificity of optogenetic manipulation, we employed localized viral injections[31]. Although this approach almost completely suppressed spontaneous activity, it spared 46% of sound-triggered firing in A2, which could have contributed to the remaining behavioral discrimination. Second, mice may perform this task by employing strategies that rely not on harmonics integration in A2, but on detection of a leading pure tone to withhold licking. The activity of pure tone-responsive primary auditory areas could allow mice to use this strategy in the absence of A2, provided that onset shifts are large enough. Regardless of this underestimation, our results indicate the contribution of A2 in defining the narrow temporal window for integrating harmonic sounds. It is important to note that the involvement of A2 in our specific behavioral task does not necessarily require its specialization for processing harmonicity. A2 could have contributed to this task using other sensory cues, such as the bandwidth of frequency distribution, dynamics in the spectral composition, or changes in sound intensity. Behavior experiments with additional controls for sensory stimuli will be needed to elucidate the range of spectro-temporal features that A2 circuits integrate.

Interestingly, acute A1 inactivation transiently abolished nearly 80% of ongoing A2 activity, which quickly recovered to reach a new equilibrium within 200 ms (Supplementary Fig. 8). This result demonstrates the hierarchical connection from A1 to A2, while also suggesting the presence of other input sources to A2 that quickly compensate for the reduction of A1 activity. This is consistent with our behavior experiments in which A1

inactivation did not significantly impair performance in harmonics discrimination. We found in our macroscopic imaging that another primary area, AAF, showed a moderate preference for harmonics (Fig. 1). Therefore, AAF may contribute to harmonics discrimination by either relaying inputs to A2 or performing harmonic integration itself. Future work will be needed to investigate the more complete response properties of all auditory cortical areas and examine whether and how spectro-temporal integration in A2 depends on multiple input sources, including A1, AAF, VAF, and thalamus.

Recent studies reported the roles of ventral auditory cortex and temporal association cortex (AuV-TeA) in innate and learned behaviors triggered by frequency-modulated sweeps or nonharmonic vocalizations[10,49]. Together with our results demonstrating preferential activation of A2, which is likely within AuV-TeA, by harmonic sounds, these temporal cortical areas may play roles in processing a broad range of natural vocalizations. Our imaging experiments showed spatial clustering of coincident harmonics-preferring neurons in a restricted area, arguing against distributed representations of harmonics in the entire AuV-TeA. At the same time, our data also showed that only 20–40% of A2 neurons displayed coincidence preference for each frequency combination (Fig. 5), suggesting heterogeneity even within this restricted area. Therefore, future work is needed to examine whether different types of complex sounds are encoded in overlapped or segregated neural populations across these areas. One interesting possibility is that the broad and heterogeneous tuning properties of A2 neurons give them the ability to integrate sounds over a broader range of spectral and temporal space than A1 neurons. AuV-TeA area may possess subnetworks that extract distinct spectro-temporal structures, including harmonics, sweeps, and vocalizations, and serve as a gateway for associating this information with behavioral relevance[50], analogous to the "ventral auditory stream" in humans and nonhuman primates[51,52].

Together, we propose that excitatory connectivity in A2 allows its neurons to integrate multifrequency sounds over a wide spectral window, whereas delayed inhibition mediated by SOM cells limits the temporal window for integration. Selective extraction of coincident multifrequency sounds in these neurons could provide neural substrates for sensory feature binding that is critical for accurately perceiving and responding to ethologically relevant sounds. New methodologies to record and manipulate activities in ensemble-specific patterns[53,54] will be instrumental in delineating the connectivity in A2 functional subnetworks, which could ultimately inform us about the neural circuits underlying vocal communication.

## Methods

**Animals**. Mice were 6–12 weeks old at the time of experiments. Mice were acquired from Jackson Laboratories: Slc32a1$^{tm2(cre)Lowl}$/J (VGAT-Cre); Pvalb$^{tm1(cre)Arbr}$/J (PV-Cre); Sst$^{tm2.1(cre)Zjh}$/J (SOM-Cre); Gt(ROSA)26Sor$^{tm9(CAG-tdTomato)Hze}$/J (Ai9); C57BL/6J; CBA/J; BALB/cJ. Both female and male animals were used and housed at 21 °C and 40% humidity with a reverse light cycle (12–12 h). All experiments were performed during their dark cycle. All procedures were approved and conducted in accordance with the Institutional Animal Care and Use Committee at the University of North Carolina at Chapel Hill and Osaka University as well as guidelines of the National Institutes of Health.

**Mouse vocalization recording**. Recordings were conducted from C57BL/6J, BALB/cJ, and CBA/J strains. Pain vocalizations were recorded in a sound isolation chamber (Gretch-Ken Industries), using custom Matlab code for recording (Mathworks) at a sampling rate of 500 kHz, microphone (4939-A-011; Brüel & Kjær), and conditioning amplifier (1708; Brüel & Kjær). The microphone was placed 5 cm above head-fixed mice. Pain vocalizations were triggered by mild electric shocks (0.5–0.8 mA, 0.5–1 s) to the tail.

**Sound stimulus**. Auditory stimuli were calculated in Matlab (Mathworks) at a sample rate of 192 kHz and delivered via a free-field electrostatic speaker (ES1; Tucker-Davis Technologies). Speakers were calibrated over a range of 2–64 kHz to give a flat response (±1 dB). For Figs. 1–6, harmonic stimuli with various F0s (2, 2.8, 4, 5.7, and 8 kHz) were generated by using harmonic components between 2 and 40 kHz. For Figs. 7 and 8, three-tone harmonic stimuli with a wider range of F0s (4, 6, 8.9, 13.4, and 20 kHz) were generated by using F0, 2 × F0, and 3 × F0. Spectrally jittered harmonics were generated by adding random frequency jitter (drawn from a uniform distribution between −0.5 × F0 and 0.5 × F0) to each component of a ten-tone harmonic sound with F0 = 4 kHz. The sound intensity for each frequency component was kept at the same sound level and was adjusted such that the total presented intensity, added in sine phases, was 75 dB Sound Pressure Level (dB SPL) (intrinsic signal imaging) or 70 dB SPL (all other experiments). Temporally shifted harmonics were generated by shifting the onset timing of their frequency components (−45 to 45 ms, 15 ms steps). For harmonic stimuli presented in Figs. 2–6, the bottom half of frequencies were shifted, and for the three-tone harmonics presented in Figs. 7 and 8, the fundamental frequency was shifted. Each frequency component had 5 ms linear rise-fall at its onset and offset. Duration of each frequency component was 100 ms for Figs. 1–7 and 300 ms for Fig. 8. Sound stimuli were presented in semi-randomized order in two-photon imaging experiments; each block of trials consisted of presentations of stimuli with all F0–Δonset (or jitter–Δonset) pairs, once each, in a randomized order, and five blocks of trials were presented. Sound frequency sets used for spectrally jittered harmonics were kept consistent within each block to allow comparison of sound representations between onset shifts. Tonal receptive fields of individual neurons were determined by presenting pure tones of 17 frequencies (log-spaced, 4–64 kHz) at 30, 50, and 70 dB SPL. For stimulation with harmonic vocalizations in intrinsic imaging experiments, a recorded sound (P100_09) from a deposited library[23] was used at 75 dB SPL mean sound intensity. Stimuli were delivered to the ear contralateral to the imaging or recording site. Auditory stimulus delivery was controlled by Bpod (Sanworks) running on Matlab.

**Intrinsic signal imaging**. Intrinsic signal images were acquired using a custom tandem lens macroscope (composed of Nikkor 35 mm 1:1.4 and 135 mm 1:2.8 lenses) and 12-bit CMOS camera (DS-1A-01M30, Dalsa). All mice were first implanted with a custom stainless-steel head bar. Mice were anesthetized with isoflourane (0.8–2%) vaporized in oxygen (1 L/min) and kept on a feedback-controlled heating pad at 34–36 °C. Muscle overlying the right auditory cortex was removed, and the head bar was secured on the skull using dental cement. For initial mapping, the brain surface was imaged through skull kept transparent by saturation with phosphate-buffered saline. For re-mapping 1–3 days before two-photon calcium imaging, the brain surface was imaged through an implanted glass window. Mice were injected subcutaneously with chlorprothixene (1.5 mg/kg) prior to imaging. Images of surface vasculature were acquired using green LED illumination (530 nm) and intrinsic signals were recorded (16 Hz) using red illumination (625 nm). Each trial consisted of 1 s baseline, followed by a sound stimulus and 30 s intertrial interval. Images of reflectance were acquired at 717 × 717 pixels (covering 2.3 × 2.3 mm²). Images during the response period (0.5–2 s from the sound onset) were averaged and divided by the average image during the baseline. Images were averaged across 10–20 trials for each sound, Gaussian filtered, and thresholded for visualization. For quantification of response amplitudes in individual areas, images were deblurred with a 2-D Gaussian window ($\sigma = 200$ mm) using the Lucy–Richardson deconvolution method[35,39]. Individual auditory areas including A1, AAF, VAF, and A2 were identified based on their characteristic tonotopic organization determined by their responses to pure tones (1 s; 75 dB SPL; 3, 10, and 30 kHz). Presentations of pure tones were interleaved with those of a harmonic vocalization, artificial harmonics (F0 = 2, 4, and 8 kHz), and/or jittered harmonics (F0 = 4 kHz, jitter drawn from a uniform distribution between −0.5 × F0 and 0.5 × F0). Regions of interest (ROIs) for quantification of response magnitudes were drawn around individual auditory areas guided by thresholded signal in individual mice. For tonotopic regions A1, VAF, and AAF, separate ROIs were drawn for 3, 10, and 30 kHz responsive areas and the response amplitudes were averaged across these three ROIs. A single ROI was drawn for A2 since the responses to three frequencies were largely overlapping.

**Two-photon calcium imaging**. Following the mapping of auditory cortical areas with intrinsic signal imaging, a craniotomy (2 × 3 mm²) was made over the auditory cortex, leaving the dura intact. Drilling was interrupted every 1–2 s and the skull was cooled with phosphate-buffered saline to prevent damage from overheating. Virus was injected at 5–10 locations (250 μm deep from the pial surface, 30 nL/site at 10 nL/min). For pyramidal cell imaging, AAV9.syn.GCaMP6s.WPRE.SV40 (2 × 10¹² genome copies per mL) was injected in VGAT-Cre×Ai9 mice. For interneuron subtype-selective imaging, AAV9.syn.Flex.GCaMP6s.WPRE.SV40 (2–4 × 10¹² genome copies per mL) was injected in either PV-Cre×Ai9 or SOM-Cre×Ai9 mice. A glass window was placed over the craniotomy and secured with dental cement. Dexamethasone (2 mg/kg) was injected prior to the craniotomy. Enrofloxacin (10 mg/kg) and meloxicam (5 mg/kg) were injected before mice were returned to their home cage. Two-photon calcium imaging was performed 2–3 weeks after chronic window implantation to ensure an appropriate level of GCaMP6s expression. Second intrinsic signal imaging was performed through the chronic window 1–3 days before calcium imaging to confirm intact auditory cortex maps. On the day of calcium imaging, awake mice were head-fixed under the two-

photon microscope within a custom-built sound-attenuating chamber. Responses to pure tones and harmonic stimuli were usually measured on two separate imaging sessions, with 53.1 ± 1.9% of cells imaged on both sessions. GCaMP6s and tdTomato were excited at 925 nm (InSight DS+, Newport), and images (512 × 512 pixels covering 620 × 620 μm²) were acquired with a commercial microscope (MOM scope, Sutter) running the Scanimage software (Vidrio) using a ×16 objective (Nikon). Images were acquired from L2/3 (200–300 μm below the surface). Lateral motion was corrected by cross-correlation-based image alignment[55]. Timings of sound delivery were aligned to the imaging frames by recording the timing of 5V trigger signals in the Wavesurfer software (Vidrio). The figure panels showing example fields of view were generated by overlaying signals from two channels using the Fiji software (https://imagej.net/Fiji).

**Surgery for in vivo electrophysiology**. After mapping auditory cortical areas with intrinsic signal imaging, the exposed skull was covered with silicone elastomer. Black cement was used during headcap implantation to prevent light exposure of the auditory cortex before recording. After 1–5 days, mice were anesthetized with isoflurane and the skull was exposed by removing the silicone cover. A small (<0.3 mm diameter) craniotomy was made above A1 or A2 based on the map obtained by intrinsic signal imaging and a durotomy was made in most experiments. Special care was taken to reduce damage to the brain tissue during this surgery, since we observe abnormal activity from damaged tissue. We found it critical to interrupt drilling every 1–2 s and cool the skull with artificial cerebrospinal fluid (aCSF, in mM: 142 NaCl, 5 KCl, 10 glucose, 10 HEPES, 2–3.1 CaCl₂, 1–1.3 MgCl₂, pH 7.4) to prevent damage from overheating.

**Extracellular unit recording with optogenetics**. For inactivation of specific interneuron subtypes, AAV9.EF1a.DIO.eNpHR3.0.EYFP.WPRE.hGH (2–5 × 10¹³ genome copies per mL) was injected into the right auditory cortex of newborn SOM-Cre or PV-Cre mice (postnatal days 1 and 2). Pups were anesthetized by hypothermia and secured in a molded platform. Virus was injected at three locations along the rostral–caudal axis of the auditory cortex. At each site, injection was performed at four depths (1000, 800, 600, and 400 μm deep from the skin surface, 23 nL/depth). Recordings were conducted 6–10 weeks after injections. On the day of recording, following a small craniotomy and durotomy in either A1 or A2, mice were head-fixed in the awake state and a 64-channel silicon probe (ASSY-77-H3, sharpened, Cambridge Neurotech) was slowly (~1 μm/s) inserted perpendicular to the brain surface. During recording, mice sat quietly (with occasional bouts of whisking and grooming) in a loosely fitted plastic tube within a sound-attenuating enclosure (Gretch-Ken Industries or custom-built). Spikes were monitored during probe insertion, and the probe was advanced until its tip reached white matter, where no spikes were observed. The reference electrode was placed at the dura above the visual cortex. The probe was allowed to settle for 1.5–2 h before collecting the data. Unit activity was amplified, digitized (RHD2164, Intan Technologies), and acquired at 20 kHz with OpenEphys system (https://open-ephys.org). A fiber-coupled LED (595 nm) was positioned 1–2 mm above the thinned skull and a small craniotomy. In interleaved trials, LED illumination was delivered that lasted from 500 ms before sound onset to 300 ms after sound offset. Since we found that excessive inactivation of inhibitory neurons causes a paradoxical elimination of sound-triggered responses and often irreversibly changes the cortical activity to a bursty state, LED intensity was kept at a moderate intensity (0.5–3 mW/mm² at the brain surface). Before starting measurements of sound-evoked responses in each mouse, we monitored spontaneous activity without sound stimuli while applying brief LED illuminations, starting from a low intensity and incrementally with higher intensities. We determined the lowest effective intensity that caused a visible increase in spontaneous activity and used one or two intensities around that level for experiments.

For area-specific silencing during extracellular recording, PV-Cre or PV-Cre×Ai9 mice were injected with AAV9.hsyn.Flex.ChrimsonR.tdTomato (3 × 10¹² genome copies per mL) 3 weeks prior to recording. Injections were made at two locations for A1 inactivation and one location for A2 inactivation (225, 450, and 720 μm deep from the pial surface, 30 nL/depth at 10 nL/min) guided by intrinsic signal imaging. A fiber-coupled LED (625 nm) was positioned 1–2 mm above the thinned skull and a small craniotomy. In interleaved trials, LED illumination (10 ms pulses at 50 Hz) was delivered that lasted 1 s before and after sound stimuli. LED intensity was around 3 mW/mm² at the surface of the brain. Pure tone responses were measured using sound stimuli at 17 frequencies (4–64 kHz, log-spaced), 70 dB SPL, and 300 ms duration. In some mice, recordings were conducted in both A1 and A2 successively in order to confirm the area specificity of optogenetic inactivation.

**In vivo whole-cell recording**. After recovery from the craniotomy and durotomy, similar to those for unit recording, mice were head-fixed in the awake state. Craniotomies were covered with aCSF and mice recovered from anesthesia for at least 1.5 h before recordings. During recording, mice sat quietly (with occasional bouts of whisking and grooming) in a loosely fitted plastic tube within a sound-attenuating enclosure (Gretch-Ken Industries). Whole-cell patch-clamp recordings were made with the blind technique. All recorded cells were located in L2/3, based on the z-axis readout of an MP-285 micromanipulator (Sutter; 130–330 μm from the pial surface). Voltage-clamp recordings were made with patch pipettes (3.8–5.7

MOhm) filled with internal solution composed of (in mM) 130 cesium gluconate, 10 HEPES, 5 TEA-Cl, 12 Na-phosphocreatine, 0.2 EGTA, 3 Mg-ATP, and 0.2 Na-GTP (7.2 pH; 310 mOsm). EPSCs were recorded at −70 mV, near the reversal potentials for inhibition, set by our internal solution. Membrane potential values were not corrected for the 15 mV liquid junction potential and series resistance was continuously monitored for stability (average $25.6 \pm 1.6$ MOhm, $n = 42$ cells). Recordings were made with a MultiClamp 700B (Molecular Devices), digitized at 10 kHz (Digidata 1440A; Molecular Devices), and acquired using the pClamp software (Molecular Devices).

**Behavioral discrimination task and optogenetic inactivation.** Water restriction was started ~1 week after area-specific adeno-associated virus (AAV) injections, which were conducted in the same way as the unit recording experiments described above. The small craniotomy for injection was filled with Dura-Gel (Cambridge NeuroTech), and the skull was coated with a thin layer of dental cement to keep its transparency. The skull was covered with silicone to prevent light access to the injected area until the testing day. Training was started 10–14 days after initiation of water restriction, when the body weight stabilized ~85% of the original weight. Mice were head-fixed in a sound isolation cubicle (Med Associates) and trained to lick for a water reward immediately following a 300 ms 4/8/12 kHz three-tone harmonics delivered to the left ear (contralateral to the injection site). The lick port was calibrated so that mice received ~5 μL water per trial. After this initial training for the simple association between the target harmonics and water reward, mice were next trained to discriminate between coincident harmonics (target) and harmonics whose fundamental frequency was shifted by 100 ms (distractor). Licking in response to the distractor was punished by a 10 s time-out period that delayed the start of the next trial. Each session lasted 100–200 rewarded trials (2–3 h), and typically only one session was conducted per day. Learning was assessed by measuring d prime for the performance on each day, and mice that reached testing criteria (d prime = 2.5) were used for testing the effect of area-specific cortical inactivation. To reduce transmission of sounds to the cortex contralateral to the optogenetic manipulation, we unilaterally presented sound either by inserting an earphone or by blocking the ear on the opposite side of the speaker with a custom-made metal bar.

On testing day, mice were challenged with smaller shifts in onset timing (0–75 ms, 15 ms steps) to determine their perceptual threshold for discrimination between coincident and shifted harmonics. A fiber-coupled LED (625 nm) was positioned 1–2 mm above the skull whose transparency was improved by adding mineral oil to the surface of the cement coating. LED intensity was ~3 mW/mm² at the surface of the skull. On ~30% of randomly interleaved trials, LED illumination (10 ms pulses at 50 Hz) was turned on 1 s before sound onset and stopped after the answer period or when the mouse made a response. Ambient red light was present in the cubicle for the duration of testing and training to prevent startling the mice when the LED was turned on. Licks were detected by breaking an infrared beam, and water delivery was controlled by a solenoid valve. The behavioral setup was controlled by Bpod (Sanworks) running on Matlab.

In a separate set of mice, area-restricted spread of AAV was confirmed by comparing the distribution of tdTomato expression with the functional map of auditory cortical areas identified by intrinsic signal imaging. In these mice, after mapping of cortical areas and viral injections, a chronic glass window was implanted, following the methods described in the two-photon calcium imaging section. Two weeks later, mice were head-fixed under the two-photon microscope, and images (512 × 512 pixels covering ~1200 × 1200 μm²) of tdTomato-expressing neurons were acquired with 925 nm excitation wavelength.

**Analysis of harmonic structures in mouse vocalizations.** Syllables were extracted from recorded vocalizations using custom Matlab codes. Syllables were detected if the absolute amplitudes exceeded 10 × standard deviation (SD) of the baseline noise. Onsets and offsets of syllables were determined using the criteria of 5 × SD, and syllables were dissected out with 15 ms margins on both ends. Contours of vocalization signals were separated from background noise by image processing of the spectrograms; the spectrogram of each syllable was produced at 1 ms temporal resolution for B6 and BALB/c and 0.5 ms temporal resolution for CBA. For each time bin, broad band components were removed by subtracting the moving average (4000 Hz window) along the frequency domain. The spectrogram was further smoothened by applying a median filter (3 × 3 pixels). The continuity of the contours was enhanced using the Frangi filter. Pixels with power >5% of the maximum value in each syllable were extracted as the signal. After extraction of the contours of vocalization signals, a flood fill algorithm was used to isolate individual continuous components within a syllable. To further reduce noise, syllable components shorter than 7 ms were rejected. Time and frequency values of individual components were obtained from the pixel positions. The F0 of harmonics was identified for each 1.4 ms segment by matching to harmonic templates. We calculated the following evaluation function (EF):

$$EF = \sqrt{\frac{1}{n}\sum_f \left(\frac{f * F0 - F_{real}}{F_{real}}\right)^2} \qquad (1)$$

where $f$ is the harmonic order of each component in the harmonic template, $n$ is the total number of harmonic components, and $F_{real}$ is the observed frequency

corresponding to the harmonic component. When the EF was <0.5, the segment was judged as harmonic and included in the analysis.

**Analysis of two-photon calcium imaging data.** Pyramidal cells were optically distinguished from GABAergic interneurons by their lack of tdTomato expression in *VGAT-Cre×Ai9* mice. ROIs corresponding to individual cell bodies were automatically detected by Suite2P software (https://github.com/cortex-lab/Suite2P) and supplemented by manual drawing. However, we did not use the analysis pipeline in Suite2P after ROI detection, since we often observed over-subtraction of background signals. All ROIs were individually inspected and edited for appropriate shapes using a custom graphical user interface in Matlab. Pixels within each ROI were averaged to create a fluorescence time series $F_{cell\_measured}(t)$. To correct for background contamination, ring-shaped background ROIs (starting at 2 pixels and ending at 8 pixels from the border of the ROI) were created around each cell ROI. From this background ROI, pixels that contained cell bodies or processes from surrounding cells (detected as the pixels that showed large increases in dF/F uncorrelated with that of the cell ROI during the entire imaging session) were excluded. The remaining pixels were averaged to create a background fluorescence time series $F_{background}(t)$. The fluorescence signal of a cell body was estimated as $F(t) = F_{cell\_measured}(t) - 0.9 \times F_{background}(t)$. To ensure robust neuropil subtraction, only cell ROIs that were at least 3% brighter than the background ROIs were included. dF/F were generated after a small offset (20 a.u.) was added to $F(t)$ in order to avoid division by extremely low baseline values in rare cases. Harmonics-evoked responses were measured as the area under the curve of baseline-subtracted dF/F traces during a 1 s window after sound onset, considering the slow kinetics of GCaMP6s. Cells were judged as significantly excited if they fulfilled two criteria: (1) dF/F had to exceed a fixed threshold value consecutively for at least 0.5 s in more than half of trials. (2) dF/F averaged across trials had to exceed a fixed threshold value consecutively for at least 0.5 s. Thresholds for excitation (pyramidal cell: 3.3 × SD during baseline period; SOM cell: 1.6 × SD; PV cell: 2.6 × SD) were determined by receiver operating characteristic analysis to yield a 90% true-positive rate in tone responses. Overall, the fraction of pyramidal cells with significant increases in the activity to at least one harmonic stimuli was 33% ($n = 336$ out of 1109 cells) in A1 and 42% ($n = 297$ out of 713 cells) in A2. Two-photon imaging fields were aligned with the intrinsic signal imaging fields by comparing blood vessel patterns and ROIs determined to be outside the areal border defined by intrinsic imaging were excluded from further analyses. CF was calculated as the frequency that triggered the strongest response at the threshold sound intensity. Some cells had sound intensity thresholds lower than our lowest tested sound intensity (30 dB). CFs in these cells were estimated as the average of two measurements: (1) frequency with strongest response at lowest sound intensity, and (2) mean frequency of a fitted Gaussian for the responses across frequencies at lowest sound intensity[35]. Bandwidth (BW70) was calculated as the average of the range of frequencies that evoked significant responses and the range of frequencies with a Gaussian fit exceeding a threshold at 70 dB SPL.

Coincidence preference was determined using mean dF/F traces across at least five trials of presentations of each sound stimulus. Normalized response magnitude was calculated for ROI–F0 (or ROI–jitter) pairs, which had significant excitatory responses in at least one Δonset. For each ROI–F0 (ROI–jitter) pair, response amplitudes were normalized to its maximum value across Δonsets, and these values were averaged across all ROIs in each cortical area. CI was calculated for each ROI–F0–Δonset (ROI–jitter–Δonset) combination only if significant excitatory responses were evoked by coincident and/or shifted harmonics. CI was calculated as $(C - S)/(C + S)$, where $C$ represents the response amplitude triggered by coincident harmonics and $S$ represents that triggered by temporally shifted harmonics. CI for onset shifts of ±15 and 30 ms in all ROIs were pooled together and compared between A1 and A2. LI in three-tone experiments was calculated for each ROI–F0–Δonset combination only if significant excitatory responses were evoked by harmonics or at least one of the component tones. LI was calculated as $(H - S)/(H + S)$, where $H$ represents the response amplitude triggered by harmonic stimulus and $S$ represents the calculated linear sum of the component tone responses. Response amplitudes were calculated as mean dF/F values during response measurement windows, and negative amplitudes were forced to 0 in order to keep the CI and LI range between −1 and 1.

Principal component analysis (PCA) was performed using Matlab's Statistics Toolbox. The ensemble activities in response to individual Δonsets were plotted in the PCA space, separately for each F0 and cortical area. Sound-evoked response amplitudes were measured as the area under the curve of baseline-subtracted dF/F traces during a 1 s window after sound onset. For each F0, normalized response vectors of individual ROIs were calculated by normalizing to the maximum response amplitudes across Δonsets. ROIs with significant excitatory responses in at least one Δonset were included. For each F0, a population response matrix of each area was made by concatenating the normalized response vectors of all responsive ROIs across all mice. For Fig. 2g, population response matrices of 7 Δonsets × 144 ROIs and 7 Δonsets × 119 ROIs were used for PCA, respectively, for A1 and A2. For Fig. 3a, a population response matrix of 9 (including 7 Δonsets as well as upper and lower components) × 74 ROIs was used for PCA. Nonsignificant responses were forced to 0 for de-noising. The same population response matrices were used for calculating pairwise Euclidean distance and correlation coefficient between ensemble activity patterns evoked by harmonics with various Δonsets. Euclidean distances were divided

by the square root of the number of ROIs to normalize for the difference in number of dimensions between A1 and A2. For each F0, single $7 \times 7$ matrix of Euclidean distance (correlation coefficient) was generated, and the values were averaged along the column, excluding the diagonal, to give the mean Euclidean distance (correlation coefficient) between each $\Delta$onset and all others. Figure 2j, k were generated by taking the average of mean Euclidean distance (correlation coefficient) traces across five F0s. Clustering of $\Delta$onset-dependent response patterns was performed using non-negative matrix factorization (NMF) with 100 repetitions. The number of clusters was determined to be three based on the Euclidean distance analysis in Fig. 2. While elbow-point analysis suggested four clusters, this simply split the negative $\Delta$onset cluster into two. For the clustering in Figs. 5 and 6, population response matrices were concatenated across both areas and all F0s (or jitters) for running NMF, then the data were split back into each F0 (jitter) and area. If a cell responded to multiple F0s (jitters) across any of $\Delta$onsets, this cell was included multiple times in this concatenated matrix, and the NMF results of these cell–sound pairs were independently assessed for clustering. Therefore, a cell could be coincidence-preferring for multiple F0s (jitters) or could be coincidence-preferring for one F0 (jitter) and shift-preferring for others. For the clustering in Supplementary Fig. 3, NMF was performed separately for each F0 (jitter) by using population response matrices concatenating A1 and A2 data. After NMF, each ROI-F0 (ROI-jitter) pair was assigned to a cluster based on the dominating coefficient. If the fraction of the dominating coefficient over the sum of all coefficients was <0.5, the ROI-F0 (ROI-jitter) pair was excluded from clusters and labeled as "others." Pairwise noise correlation between ROIs was calculated as the Pearson's correlation coefficient between response vectors concatenating mean-subtracted response amplitudes across all trials and sound stimuli. Noise correlation was calculated between simultaneously recorded ROIs in each mouse, and the values were concatenated across mice. To control for the physical distance-dependence of noise correlation, each same-F0 cell pair was assigned three distance-matched random cell pairs drawn from all the ROI pairs within the same field of view. To account for the dependence of noise correlation on imaging sensitivity, two separate controls were taken: noise level and SNR. Noise level of each ROI was calculated as the SD of dF/F trace during baseline period without sound stimuli. SNR of each ROI was calculated as the peak sound-evoked response amplitude divided by SD. For each ROI, two random ROIs with the closest SD (SNR) were drawn from the same field of view. Thus, each same-F0 cell pair had four SD (SNR)-matched control cell pairs: combinations between $2 \times 2$ control ROI sets.

**Analysis of in vivo whole-cell recording data**. Data were analyzed using custom programs in Matlab. Response amplitudes were quantified from traces averaged across 3–9 trials of each sound stimulus. For three-tone harmonics responses, EPSC was measured as negative going charge during the entire sound duration in order to capture onset responses from both leading and lagging sounds. Latency of onset responses was taken to be 10 ms. LI was calculated for the F0 that evoked the strongest responses in each cell. LI was calculated as $(H - S)/(H + S)$, where $H$ represents the response amplitude triggered by harmonic stimulus and $S$ represents the calculated linear sum of the component tone responses. LI for onset shifts of ±15 and 30 ms were averaged together and this value was compared with LI for coincident harmonics.

**Analysis of multiunit recording data**. Single- and multiunits were isolated using Kilosort software (https://github.com/cortex-lab/KiloSort) and spike-sorting graphical user interface (Phy; https://github.com/cortex-lab/phy). Positions of cortical surface, layers, and white matter were identified by current source density analysis and the distribution of multiunit spikes. Multiunit spikes were calculated by combining all the spikes within each channel (Fig. 4), layer (Fig. 8), or across layers (Supplementary Fig. 8). For interneuron inactivation experiments (Fig. 4 and Supplementary Fig. 2), peristimulus time histogram (PSTH) was generated at 10 ms bins and baseline firing rate 0–200 ms before sound onsets was subtracted. For visualization in the figures, PSTHs were smoothened by three bins moving average. Harmonics-evoked responses were quantified as the sum of positive-going PSTH during 0–150 ms after sound onset. Harmonics response magnitudes were normalized to the maximum response magnitude in the control trials in individual units. Units in superficial layers (L2/3 and L4) were included in the analysis. For cortical inactivation experiments (Fig. 8e, f and Supplementary Fig. 8), PSTH was generated at 10 ms bins and normalized to the spontaneous firing rate 0–100 ms before LED onset. Optogenetic suppression of spontaneous firing was quantified as the normalized spike frequency 800–1000 ms after LED onset. Sound-evoked responses were quantified as the sum of positive-going PSTH during sound duration, after subtraction of the baseline firing rate 0–200 ms before sound onsets. Thus, sound-evoked responses do not include the decrease in the spontaneous firing rate caused by LED. For detailed analysis of the suppression kinetics at the LED onset (Supplementary Fig. 8), PSTH was generated at 0.5 ms bin and normalized to the spontaneous firing rate of 0–10 ms before LED onset. Suppression latency and tau were obtained by fitting a single-exponential curve to the PSTH.

**Behavioral analysis**. To quantify the effect of cortical inactivation, psychometric curves describing response rate against $\Delta$onset were drawn separately for control and LED trials in each mouse. $\Delta$onsets were converted to log-modulus scale to facilitate fitting of the psychometric curve to a logistic function. The sigmoid's midpoint (behavioral threshold) and the logistic growth rate ("slope") were fit to the curve. Trials at the beginning and the end of each session where mice were not motivated to lick were excluded from the analysis. Only the first 250 trials on test day in each mouse were analyzed to quantify the optogenetic effect, since we observed a gradual improvement in behavioral performance in photostimulation trials, which likely reflects an adaptation of behavioral strategy over the course of repeated optogenetic manipulation[56]. Nevertheless, the results held true even if we included all trials (Supplementary Fig. 9d, e).

**Statistical analysis**. All data are presented as mean ± SEM. Statistically significant differences between conditions were determined using standard parametric or nonparametric tests in Matlab. Two-sided paired $t$ test or Wilcoxon's signed-rank test were used for paired tests, and Wilcoxon's rank-sum test was used for independent group comparisons. For comparison of multiple groups, one- or two-way analysis of variance was followed by Tukey's honest significance test. In cases where parametric statistics are reported, data distribution was assumed to be normal, but this was not formally tested. Randomization is not relevant for this study because there were no animal treatment groups. All $n$ values refer to the number of cells, except when explicitly stated that the $n$ is referring to the number of mice or number of cell–sound pairs. Behavioral training was conducted blind to the virus injection site in individual mice. However, blinding was not possible for behavioral testing since photostimulation needed to be aimed at the appropriate target region. Nonetheless, optogenetic manipulation was performed in randomly interleaved trials, which ensured the appropriate internal control within each animal. Sample sizes were not predetermined by statistical methods, but were based on those commonly used in the field.

**Reporting summary**. Further information on research design is available in the Nature Research Reporting Summary linked to this article.

## Data availability
The data that support the findings of this study are available from the corresponding authors upon reasonable request. Source data are provided with this paper.

## Code availability
Custom Matlab codes used in this study will be made available from the corresponding author upon reasonable request.

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

## Acknowledgements

We thank Paul Manis, Toshihide Hige, and Jose Rodriguez-Romaguera for advice throughout the project and comments on the manuscript, and Taha Lodhi for help in behavioral training. This work was supported by NIDCD (R01DC017516), Pew Biomedical Scholarship, Whitehall Foundation, Klingenstein-Simons Fellowship (H.K.K.), Foundation of Hope (H.K.K. and H.T.), and NINDS (F31-NS111849, T32-NS007431; A.M.K.).

## Author contributions

A.M.K. and H.K.K. designed the project. A.M.K., D.A.A., and H.K.K. conducted in vivo experiments and analyzed data. A.G. helped with clustering data analyses. H.T. performed the vocal recordings and analysis. A.M.K. and H.K.K. wrote the manuscript, with inputs from other authors.

## Competing interests

The authors declare no competing interests.
