## [Peer Review File · Nature Communications]

REVIEWER COMMENTS

Reviewer #1 (Remarks to the Author):

The authors present a timely and highly original body of work indicating that mouse auditory cortex (AC) contains a population of functionally connected neurons, whose activity is selective for broadband sounds with coincident spectral components. The study is motivated by asking how neurons in AC might bind together harmonic frequency components of a sound into a unified percept, as must be done for speech communication. The authors propose that the selectivity for coincident spectral components in a specialized population of neurons in A2 might explain perceptual binding. The authors make thorough use of state-of-the-art techniques in neuroscience. Importantly, they provide functional roles for SOM vs PV neuron inhibitory control of temporal integration in excitatory neurons, which is an important new contribution to our understanding of neural circuitry. Since the neural basis of sensory feature binding is generally poorly understood, this study will bring interest from across the wider field. Appropriate statistical tests are used, and except where noted below, enough detail is given for reproducibility.

A primary criticism is that the authors focus on the role of spectral harmonicity because of its relationship to vocalizations, however, most of the data suggest that spectral coincidence detection in A2 occurs for both harmonic and inharmonic sounds. Second, the evidence causally relating activity in A2 to perceptual binding as a task-relevant cue requires better explanation to interpret the results beyond an ambiguous change in behavioral response rate.

Major Comments:

- 1) The authors begin by showing how different regions of AC respond to harmonic vocalizations. Based on those results, the authors focus on the uniqueness of A2 vs A1 for the remainder of the paper, presumably to simplify presentation. However, figures 1e-h show that a gradient of preference exists across different regions of AC. Thus, to present A2 as unique without having compared against AAF, for example, might be misleading in terms of A2 being a specialized locus of spectral coincidence processing.
- 2) It would be good to contrast an 'Inharmonic/tone ratio' vs the 'Harmonic/tone ratio', to clarify if regions of AC are truly selective for harmonic sounds, or simply respond better to broadband sounds than pure tones.
- 3) It is not clear in the main text how the data were processed before PCA analysis. Which features were extracted, e.g. the mean or max across a time window per trial? Also, it is not clear what was correlated to construct figure 2i and related figures. Thus, I'm not sure how to interpret those figures.
- 4) I could not understand what is being plotted in the "Fraction of overlap" figures, e.g., figure 5d. What does 'overlap' mean? What is being counted?
- 5) The section title, 'A2 subnetworks exist only for sounds with harmonic regularity', is not correct, since figure 6i shows that the noise correlations for inharmonic sounds were significantly larger than randomly selected distanced-matched cells. Thus, A2 subnetworks exist for sounds with harmonic irregularity.
- 6) In general, it seems that the authors' evidence for the importance of harmonicity to neurons in A2 rests mostly on the noise correlation analysis, since coincidence detection results were similar for harmonic and inharmonic sounds. However, it is not clear how the data were clustered for this analysis. It would be good to separate inharmonic and harmonic cells in figure 6d. Also, how were the harmonic and jittered cells selected? In other words, are the 'Harmonic' cells those that somehow 'preferred' harmonic sounds? Are the harmonic-cluster and jitter-cluster entirely different populations?
- 7) The authors state that neuronal networks are stable across days, however, no evidence is shown that the same cells were in fact identified across days, which is important since the authors indicate

that only ~50% of neurons were visible across multiple sessions.

8) Finally, the authors' paradigm for optogenetic manipulation of behavior is a powerful addition to the manuscript that indicates a causal role for A2 in their task. However, the data are difficult to interpret as a behavioral impairment because 'Response Rate' is not defined as false alarm (or potentially correct reject?) in figure 8g or the text, and it is not clear how to contextualize supp. Figure 8d (which should be included in the main text).

Minor comments:

- 1) Please include stimulus bandwidths in the main text.
- 2) Line 201: It is not clear what is meant by, "distinct but overlapping".
- 3) Line 317: It is not clear what is meant by "this simple but critical sound feature".
- 4) Line 336: Please clarify how harmonics-specific subnetworks potentially contribute to perception through computations such as pattern completion.
- 5) Line 339: Cite Hebb.
- 6) Line 364: It is not immediately apparent to me that perceptual phenomena operating on millisecond vs seconds-long timescales are "likely" to use the same mechanisms.
- 7) Since some strains of mice are known to have early-onset hearing loss, please indicate (1) the maximum age of the mice, and (2) that their hearing was normal at the time of the experiments.
- 8) Please provide more detail about the custom tandem lens used for widefield imaging.
- 9) I think that figure 2 shows distinct neuronal ensemble response patterns, not distinct ensemble members. If so, please change the figure title.
- 10) CI and LI are not defined in the figure captions.
- 11) 'Overlap' figure panels need an explanatory title.

Reviewer #2 (Remarks to the Author):

In this manuscript, the authors investigate the role of secondary auditory cortex (A2) in harmonic processing (specifically the sensitivity to the synchronous timing of harmonics) using behaviour, electrophysiology, 2-photon microscopy, and optogenetics. This is without a doubt a very impressive set of data, studying a very interesting yet understudied aspect of auditory neuroscience.

My main criticism relates to how the data is interpreted. Harmonic complexes are difficult to study as there are a high number of features a neuron may be tuned to (one or more particular frequencies that make up the sound and/or their harmonic spacing). Without studying the tuning of each neuron to each individual harmonic, in addition to two tone combinations in relation to its response to the more complex stimuli, it is not clear whether a neuron's response can be explained by simpler mechanisms. These may look like harmonicity tuning for a specific stimulus, but may not generalize to other stimuli. For example, a neuron may be a multiplex neuron (identified in many previous studies), tuned to one low and one high frequency tone. Playing both tones together leads to supralinear summation and a significantly higher firing rate. Can your observations be explained by such neurons or is something else required? Are neurons responding to all the harmonics in your stimulus, or just two tones? Assuming it is only two tones, does the frequency separation between these two tones matter? A more parsimonious interpretation is that A2 neurons are integrating multiple frequency components, with perhaps a subset of these neurons are tuned to harmonically related inputs. While the noise correlation data suggest that harmonic stimuli may be processed differently than inharmonic stimuli, many A2 neurons don't seem to differentiate (Extended data figure 3). And any task or stimulus requiring wideband integration, not just detecting the synchronous onset of harmonics, may benefit from the response properties of A2 neurons. From the data shown in Fig. 2, while A2 neurons have a higher coincidence preference than A1 neurons, many neurons still have no coincidence preference (or even a negative preference), suggesting that this region is more heterogeneous than just being the "a locus for harmonic integration" as mentioned in the abstract. To be clear, I am not suggesting any additional experiments, rather a suggestion to tone down your main claims. It is my opinion that it

would be more reasonable to say that A2 is involved in processing sounds over a wider spectral and temporal integration window than A1, which is likely advantageous for harmonic and vocalization processing.

Minor comments

1. figure 7f-it is difficult to see the small changes between the observed and linear sum. I would recommend having an inset plot for one trace, where you make your point more clearly. Looking at the individual plots, it does seem that some responses are not linear early on, when just the single tone is played, but how can this be the case if the other two tones haven't been played yet...

2. Fig 7: It is hard to interpret the result that the summation is always sublinear, but just less so in A2 (so more linear)...Is this just the consequence of SOM neurons making the response sublinear?

3. Line 305-" these results indicate that A2 is causally related to perceptual discrimination of harmonic sounds..."

The behavioural data does not demonstrate the importance of harmonic sounds as harmonicity is not varied. Furthermore, multiple features are changing between stimuli (frequencies static or dynamic, sound level differences, duration, etc.) and no control experiments have been run to demonstrate what acoustic features the animal is using. This is perhaps the weakest part of an otherwise very strong manuscript. I would suggest leading with the behavioural experiment, which is at best suggestive that A2 is doing something different than A1, akin to wideband spectral processing, and then following up with understanding the properties of A2 (harmonic vs. inharmonic), and only then mechanistic issues such as the role of inhibition.

4. A note on extended data figure 8- positive onsets may have not worked due to combination tones produced at F_0 by $2F_0$ and $3F_0$ played together (masking the sudden onset of F_0 when delayed, as the sound was already present as a combination tone)

Reviewer #3 (Remarks to the Author):

In this manuscript, Kline and colleagues have explored the neural circuit mechanisms underlying perceptual binding of coincident harmonic sounds. First, they localized A2 as a locus for processing harmonics. They found that neurons in A2, but not in A1, are particularly sensitive to shifts in component timings. Moreover, inactivating SOM, but not PV neurons, expanded the temporal window of integration, revealing a local neural circuit mechanism that shapes harmonic processing. Finally, perturbing A2 activity resulted in a shallower response rate slope in a Go-No Go task that required discrimination of coincident harmonics.

Overall, the study is timely, interesting and well-executed. The experiments are expertly performed and the manuscript is well written. Comparing and contrasting neural activity and perturbations in A2 vs. A1 is a major strength of the study. I support the publication of this manuscript in Nature Communications.

Here are a few suggestions to improve the manuscript –

1. Using PCA, Euclidean distance and correlation measures in Figure 2g-i, the authors claim that A1 neuronal representation changes continuously with time onset, consistent with a gain-control model, whereas A2 representation shows more abrupt transitions. However, using NMF in 5a, the authors find three distinct clusters in both A1 and A2. It is unclear why that is so. Authors should clarify the reason for this discrepancy. I am convinced that there are more neurons in A2 for coincidence detection compared to A1 (Figure 5b) but the evidence for gain control in A1 vs. non-overlapping representation in A2 is indirect at best. It would help to provide more direct analyses to convincingly prove this point. For example, does scaling (consistent with gain control) the response profiles of

individual neurons in A1 for zero delta onset (for e.g., in Fig 2c) adequately describe the responses at other delta onsets?

2. Inferring functional subnetworks using noise correlation analyses in Figure 5f-g and Figure 6i, is the weakest part of the paper. Noise correlation analysis is meant to capture the correlated fluctuations of neuronal firing rates between simultaneously recorded neurons and is, at best, an indirect surrogate for ground-truth connectivity. I have several methodological concerns about this analysis. First, estimating noise correlations using calcium imaging is especially challenging because of temporal smoothing of spiking activity that obscures the co-fluctuations that one is trying to measure. Second, it is unclear from the method section whether this analysis was done on simultaneously recorded neurons (which should be the case) or on a pseudopopulation of concatenated neurons across multiple sessions. Third, noise correlations are very sensitive to the SNR of the calcium traces and adequate controls need to be performed. Fourth, the difference in noise correlations between A1 and A2 is statistically significant (Figure 5g) but too subtle to allay my concerns. Importantly, the major results of the paper about A2 neural activity, circuit mechanism and behavioral role still hold without claiming that “Therefore, while A2 integrates coincident sounds regardless of spectral structure, it has specialized excitatory subnetworks dedicated to binding harmonics.” In my opinion, the sections on inferring excitatory subnetworks using noise correlations distracts from the main results.

3. A major result of this study is the contribution of A2 (but not A1) in perceptual discrimination of harmonics (Figure 8). Are the psychometric curves plotted in Figure 8g from an example animal? Was there any effect on reaction times after A1 and A2 inactivation? Authors should also include a supplementary figure panel to show the individual psychometric curves for each animal to give a sense of behavioral variability and the effect of optogenetic silencing, in addition to the half-max delta onset summary in Figure 8h. Instead of focusing on a point estimate of delta onset at half-max performance, the authors could try to include all data points and quantify the slope of the individual response curves. The expectation is that the distribution of slopes will be shallower for A2 inactivation compared to control.

Minor points:

1. Can the authors provide some justification for the parameter choices (highlighted below) for the 2-photon imaging response calculation pipeline? This is important given that all subsequent analyses depend upon this step. What fraction of neurons end up being significantly modulated based on these choices? Please include this number either in the main text or in the figure 2 legend.

The remaining pixels were averaged to create a background fluorescence time series $F_{\text{background}}(t)$. The fluorescence signal of a cell body was estimated as $F(t) = F_{\text{cell_measured}}(t) - 0.9 \times F_{\text{background}}(t)$. To ensure robust neuropil subtraction, only cell ROIs that were at least 3% brighter than the background ROIs were included. Normalized time series dF/F were generated after a small offset (20 a.u.) was added to $F(t)$ in order to avoid division by extremely low baseline values in rare cases. Harmonics-evoked responses were measured during 1-s window after sound onsets. Cells were judged as significantly excited if they fulfilled two criteria: 1) dF/F had to exceed a fixed threshold value consecutively for at least 0.5 s in more than half of trials. 2) dF/F averaged across trials had to exceed a fixed threshold value consecutively for at least 0.5 s.

2. Single unit example in Fig 4b is not very convincing because the control responses (black traces) does not decay much at non-zero delta onsets, making it hard to see the difference after SOM inactivation.

3. Collapsing across all delta onsets to plot the cumulative distributions in Fig 4m and 4p obscures the effect, which is predominant at small deviations from 0 (Fig 4o).

4. On page 9 first paragraph, the authors suggest that “similar to pyramidal cells, PV cells exhibited strong coincidence preference in A2, but less in A1 (Fig. 4o, p; A1: 0.066 ± 0.003 ; A2: 0.171 ± 0.006 ;

A1 vs A2: $p = 0.0001$), suggesting that they follow the response properties of surrounding pyramidal cells.” Given these experiments, it is impossible to know if the PV neurons follow the pyramidal cells or vice versa.

5. Figure 8e should include electrophysiology data examples from A2 in addition to A1, since that is the focus of the study.

Response to Reviewers:

We are delighted that all three reviewers appreciated our original submission and found that *“The authors present a timely and highly original body of work”, “this study will bring interest from across the wider field”, “This is without a doubt a very impressive set of data, studying a very interesting yet understudied aspect of auditory neuroscience”, and “Overall, the study is timely, interesting and well-executed. The experiments are expertly performed and the manuscript is well written.”* We appreciate all the constructive suggestions that helped us significantly improve our manuscript. We have performed additional experiments and data analyses, incorporated new figures, and revised the text as suggested by the reviewers. Below, please find our responses to all of the specific comments made by the reviewers. Major changes are highlighted in red in the revised manuscript.

Reviewer 1:

We thank Reviewer 1 for their kind words regarding our manuscript. We are especially delighted that this reviewer found our work to be a *“timely and highly original body of work”* and stated that our finding on the functional roles of SOM and PV inhibitory neurons is *“an important new contribution to our understanding of neural circuitry.”* Below are the responses to this reviewer’s comments.

Major Comments

1) The authors begin by showing how different regions of AC respond to harmonic vocalizations. Based on those results, the authors focus on the uniqueness of A2 vs A1 for the remainder of the paper, presumably to simplify presentation. However, figures 1e-h show that a gradient of preference exists across different regions of AC. Thus, to present A2 as unique without having compared against AAF, for example, might be misleading in terms of A2 being a specialized locus of spectral coincidence processing.

We agree with the reviewer that there is a gradient of preference across auditory cortical areas and that our description of A2 as “unique” could therefore be misleading. To address this, we first considered the possibility that some of the signals in AAF could have been contaminated from A2, considering their proximity in some animals (e.g. Fig. 2b). To reduce cross-contamination, we adopted the deblurring used in previous studies to compensate for the scattering of photons within tissue (Issa et al., *Neuron* 2014 doi: 10.1016/j.neuron.2014.07.009; Romero et al., *Cereb. Cortex* 2020 doi: 10.1093/cercor/bhz190) and updated all the quantification of our intrinsic signal imaging data (Fig. 1e-h, Extended Data Fig. 1, and newly added Fig. 6a). As seen in the plots, deblurring enhanced the overall difference between A2 and AAF; however, AAF still showed an intermediate level of harmonics preference, which was significantly higher than that of A1. Therefore, to avoid giving a potentially incorrect impression that A2 is the only area responsible for multi-frequency integration, we revised the text

throughout the Results and Discussion to soften our claim. First, we updated the results section to say:

“Measuring the ratio of vocalization to pure tone responses highlighted the inter-area difference, with the strongest contrast observed between A1 and A2 (A1: 0.99 ± 0.04 ; A2: 1.96 ± 0.09 ; $p < 0.0001$), and AAF being intermediate (1.51 ± 0.09 ; A1 vs AAF: $p < 0.0001$; A2 vs AAF: $p < 0.0001$).”

We also acknowledged AAF’s potential role in harmonics integration in the Discussion as:

“We found in our macroscopic imaging that another primary area, AAF, showed a moderate preference for harmonics (Fig. 1). Therefore, AAF may contribute to harmonics discrimination by either relaying inputs to A2 or performing harmonic integration itself. Future work will be needed to investigate the more complete response properties of all auditory cortical areas and examine whether and how spectro-temporal integration in A2 depends on multiple input sources, including A1, AAF, VAF, and thalamus.”

Furthermore, throughout the manuscript, we removed words such as “specialized” or “unique” in regards to A2 as a site for harmonics preference and multi-frequency integration.

2) It would be good to contrast an ‘Inharmonic/tone ratio’ vs the ‘Harmonic/tone ratio’, to clarify if regions of AC are truly selective for harmonic sounds, or simply respond better to broadband sounds than pure tones.

We thank the reviewer for this idea to enhance our manuscript. We have performed additional intrinsic imaging experiments using both inharmonic and harmonic sounds in the same mice and found that A2 shows preference to both harmonic and inharmonic sounds over pure tones, consistent with our two-photon calcium imaging data. This data is now included as new panels in Fig. 6a and Extended Data Fig. 6a and is described in Results as:

“we next asked if the functional subnetworks we identified in A2 are specific to harmonic sounds or generalized to inharmonic sounds. To answer this, we introduced spectral jitters to ten-tone harmonics (4-kHz F0, Fig. 6a) and examined how they affect cortical harmonic representations. We first conducted intrinsic signal imaging of all auditory cortical areas using both harmonics and jittered sounds and found that both sounds preferentially activated A2 (Fig 6a; Extended Data Fig. 6a; multi-freq/tone ratio, harmonic: 1.94 ± 0.13 ; jittered: 1.79 ± 0.29 , $p = 0.278$).”

3) It is not clear in the main text how the data were processed before PCA analysis. Which features were extracted, e.g. the mean or max across a time window per trial? Also, it is not clear what was correlated to construct figure 2i and related figures. Thus, I’m not sure how to interpret those figures.

We apologize for the lack of clarity in the description of our analysis methods. The quantification methods were described in our original Methods, but we now see that they were hard to

understand since they were spread out across subsections. We have reorganized the 'Analysis of two-photon calcium imaging data' section of Methods and added more detail to the PCA and correlation coefficient analysis methods as:

“Principal component analysis (PCA) was performed using Matlab’s Statistics Toolbox. The ensemble activities in response to individual Δ onsets were plotted in the PCA space, separately for each F0 and cortical area. Sound-evoked response amplitudes were measured as the area under the curve of baseline-subtracted dF/F traces during a 1-s window after sound onset. For each F0, normalized response vectors of individual ROIs were calculated by normalizing to the maximum response amplitudes across Δ onsets. ROIs with significant excitatory responses in at least one Δ onset were included. For each F0, a population response matrix of each area was made by concatenating the normalized response vectors of all responsive ROIs across all mice. For Fig. 2g, population response matrices of 7 Δ onsets \times 144 ROIs and 7 Δ onsets \times 119 ROIs were used for PCA, respectively, for A1 and A2. For Fig. 3a, a population response matrix of 9 (including 7 Δ onsets as well as upper and lower components) \times 74 ROIs was used for PCA. Non-significant responses were forced to 0 for de-noising. The same population response matrices were used for calculating pairwise Euclidean distance and correlation coefficient between ensemble activity patterns evoked by harmonics with various Δ onsets. Euclidean distances were divided by the square root of the number of ROIs to normalize for the difference in number of dimensions between A1 and A2. For each F0, single 7 \times 7 matrix of Euclidean distance (correlation coefficient) was generated, and the values were averaged along the column, excluding the diagonal, to give the mean Euclidean distance (correlation coefficient) between each Δ onset and all others. Fig. 2j, k were generated by taking the average of mean Euclidean distance (correlation coefficient) traces across five F0s. “

4) I could not understand what is being plotted in the “Fraction of overlap” figures, e.g., figure 5d. What does ‘overlap’ mean? What is being counted?

We thank the reviewer for highlighting their confusion with regards to this figure. We have updated Fig. 5d to include the title “Overlap between coincidence-pref clusters across F0s” and inserted a description in the legend as “observed overlap count (that is, cells counted in the coincidence-preferring cluster for multiple F0s)”, which we hope increases clarity for this data. This figure shows that the observed frequency of the same cells counted as coincidence-preferring for multiple F0s is above the chance level predicted from shuffled data.

5) The section title, ‘A2 subnetworks exist only for sounds with harmonic regularity’, is not correct, since figure 6i shows that the noise correlations for inharmonic sounds were significantly larger than randomly selected distanced-matched cells. Thus, A2 subnetworks exist for sounds with harmonic irregularity.

We agree with the reviewer that this sentence is an overstatement that ignores the potential formation of subnetworks by inharmonics-responding neurons. We have revised the section title to read,

“A2 subnetworks are preferentially recruited by sounds with harmonic regularity“

to more accurately describe our results. Additionally, we have revised the Results section to state,

“We note that the noise correlation between the neurons responding to jittered sounds was significantly higher than between distance-matched random cell pairs ($p = 0.002$). The above-chance overlap between coincidence-preferring cells for harmonic and jittered sounds (Fig. 6h) likely accounts for this observation and may suggest that jittered harmonics recruit harmonics-encoding subnetworks through a pattern completion mechanism in A2 circuits.”

Finally, we removed other instances throughout the paper where we refer to “specialized” excitatory subnetworks within A2 and use the phrase “harmonics-preferring” rather than “harmonics-specific.”

6) In general, it seems that the authors’ evidence for the importance of harmonicity to neurons in A2 rests mostly on the noise correlation analysis, since coincidence detection results were similar for harmonic and inharmonic sounds. However, it is not clear how the data were clustered for this analysis. It would be good to separate inharmonic and harmonic cells in figure 6d. Also, how were the harmonic and jittered cells selected? In other words, are the ‘Harmonic’ cells those that somehow ‘preferred’ harmonic sounds? Are the harmonic-cluster and jitter-cluster entirely different populations?

We believe this confusion arose partly due to the poor definition of “population response matrix” in our original Methods, which we have addressed by rewriting the description for PCA visualization in response to comment 3) above. To address this reviewer’s specific questions, we further elaborated our Methods section by inserting a description as:

“For the clustering in Figures 5 and 6, population response matrices were concatenated across both areas and all F0s (or jitters) for running NMF, then the data were split back into each F0 (jitter) and area. If a cell responded to multiple F0s (jitters) across any of Δ onsets, this cell was included multiple times in this concatenated matrix, and the NMF results of these cell-sound pairs were independently assessed for clustering. Therefore, a cell could be coincidence-preferring for multiple F0s (jitters) or could be coincidence-preferring for one F0 (jitter) and shift-preferring for others.”

We hope this answers the questions raised by this reviewer. “Harmonic” and “Jittered” cells in Fig. 6f refer to all the cells that showed responses to at least one Δ onset for harmonic and jittered sounds, respectively. Out of these responsive neurons, 25.0% (harmonic) and 27.2% (jittered) were judged as coincidence-preferring in A2. Since they were judged independently, the same cell could be included in coincidence-preferring clusters for both harmonic and jittered

sounds. Indeed, our Fig. 6h (originally Fig. 6g) shows that there is above-chance frequency where a cell is included in coincidence-preferring clusters for both harmonic and jittered sounds.

Our Extended Data Fig. 3b shows clustering data separately for harmonic and jittered sounds, which shows a similar abundance of the coincidence-preferring cluster in both sounds.

7) The authors state that neuronal networks are stable across days, however, no evidence is shown that the same cells were in fact identified across days, which is important since the authors indicate that only ~50% of neurons were visible across multiple sessions.

We thank the reviewer for pointing this out, and we believe our further analysis (Extended Data Fig. 4e) on this data is a strong addition to our manuscript. In this new analysis, we focused on only the Same-F0 cell pairs imaged on both days. Noise correlation was measured for identical Same-F0 cell pairs across two days (harmonics and pure tone experiments) and plotted against each other. We observed a positive correlation across days ($n = 84$ cell pairs, $R = 0.421$, $p < 0.0001$). Furthermore, 68.3% of the data points fell in the upper right quadrant, indicating positive noise correlation (and functional connectivity) on both days. We have addressed this in the Results section by stating:

“When noise correlation in individual same-F0 cell pairs was compared between harmonics and pure tone experiments, these values showed a significant positive correlation with each other, supporting an overlapping set of cells in the subnetwork across days (Extended Data Fig. 4e).”

8) Finally, the authors' paradigm for optogenetic manipulation of behavior is a powerful addition to the manuscript that indicates a causal role for A2 in their task. However, the data are difficult to interpret as a behavioral impairment because 'Response Rate' is not defined as false alarm (or potentially correct reject?) in figure 8g or the text, and it is not clear how to contextualize supp. Figure 8d (which should be included in the main text).

We defined response rate as trials in which the mice licked the water port divided by total trials for each Δ onset. Thus, it indicates a hit rate for the target sound and a false alarm rate for distractors. To increase clarity, we renamed “Response Rate” as “Hit + FA Rate” in Fig. 8g.

Following the reviewer's suggestion, we have moved former Extended Data Fig. 8d to the main figure, now Fig. 8h. This panel shows that the LED did not affect the ability of the mice to respond to the target. Rather, A2 inactivation affected their behavior via an increase in FA responses to the shifted harmonics. We suspect that our mice err by increasing FA rather than decreasing Hit due to our training paradigm. Since we first train our mice with the easy sound-reward association and then proceed to the challenging discrimination task (which is a common strategy in training mice without discouraging them), the default response of our mice is to lick, and they learn to withhold licking in response to distractors. Indeed, the longest part of the training is for them to learn to withhold licking to the distractors. Therefore, when A2 inactivation confuses the mice, they are more likely to follow their default behavior and respond with licking.

Minor comments:

1) Please include stimulus bandwidths in the main text.

This information was in the Methods, but we have amended the results section for clarification as:

“To examine if preferential activation of A2 is specific to this vocalization or generalized to harmonic structures, we next presented artificially generated harmonics ($F_0 = 2, 4, 8$ kHz, harmonic stacks up to 40 kHz)”.

2) Line 201: It is not clear what is meant by, “distinct but overlapping”.

We have changed this sentence to read:

“Coincident harmonics-preferring neurons for five F_0 s were spatially intermingled with each other and represented partially overlapping populations in A2 (Fig. 5c, d).”

3) Line 317: It is not clear what is meant by “this simple but critical sound feature”.

We meant that multi-frequency sounds are a relatively simple sound feature on their own, but that the integration of these sounds is critical for our perception of more complicated sounds such as language. We have changed this sentence to read:

“Integration of coincident multi-frequency sounds, a simple and prevalent sound feature, could be fundamental for the perception of more complicated sounds, such as language.”

4) Line 336: Please clarify how harmonics-specific subnetworks potentially contribute to perception through computations such as pattern completion.

We have revised this sentence to read:

“Noise correlation analyses revealed harmonics-preferring subnetworks within A2, which potentially contribute to perception through computations such as gamma oscillations and pattern completion (for example, our ability to fill in missing fundamentals).”

5) Line 339: Cite Hebb.

We inserted a citation to Hebb, D. O. The organization of behavior: a neuropsychological theory. (Wiley Interscience, 1949).

6) Line 364: It is not immediately apparent to me that perceptual phenomena operating on millisecond vs seconds-long timescales are “likely” to use the same mechanisms.

We agree that it is unlikely that the delayed inhibition alone can account for the sound stream segregation in general. However, we think we cannot avoid discussing the temporal coherence theory by Shamma et al. (Trends Neurosci. 2011) since they proposed a hypothesis that delayed inhibition accounts for a specific case of stream segregation of alternating tones. We have softened this sentence by rewriting to:

“Although the temporal coherence theory was proposed for sound stream segregation which evolves over larger time scales (seconds), its mechanisms could partially share properties with those of the concurrent (single-presentation) integration/segregation investigated in this study.”

7) Since some strains of mice are known to have early-onset hearing loss, please indicate (1) the maximum age of the mice, and (2) that their hearing was normal at the time of the experiments.

For all imaging and electrophysiology experiments, mice were between 6 and 12 weeks old at the time of the experiment. For all mice used for two-photon calcium imaging and electrophysiology experiments, we performed intrinsic signal imaging the day before imaging or recording to ensure that their auditory cortex maps were intact (displayed 3, 10, and 30 kHz responses). Therefore, we believe that the hearing of our mice was normal around our stimulus frequency range (2-40 kHz) at the time of the experiments.

8) Please provide more detail about the custom tandem lens used for widefield imaging.

We included a description of the lenses as “a custom tandem lens macroscope (composed of Nikkor 35 mm 1:1.4 and 135 mm 1:2.8 lenses)” in the Methods.

9) I think that figure 2 shows distinct neuronal ensemble response patterns, not distinct ensemble members. If so, please change the figure title.

We changed the figure title to “Coincident and shifted harmonics recruit distinct neuronal ensemble patterns in A2.”

10) CI and LI are not defined in the figure captions.

CI and LI are now defined as “Coincidence-preference index” and “Linearity index” in the legends of Figure 2 and Figure 7, respectively.

11) ‘Overlap’ figure panels need an explanatory title.

We added titles above the panels in Fig. 5d and Fig. 6h.

Reviewer 2

We thank Reviewer 2 for their helpful suggestions for improving our manuscript, and we are especially pleased that this reviewer thought that we present “*a very impressive set of data, studying a very interesting yet understudied aspect of auditory neuroscience*”. Please see our comments to the reviewer below.

Major Comments:

Harmonic complexes are difficult to study as there are a high number of features a neuron may be tuned to (one or more particular frequencies that make up the sound and/or their harmonic spacing). Without studying the tuning of each neuron to each individual harmonic, in addition to two tone combinations in relation to its response to the more complex stimuli, it is not clear whether a neuron’s response can be explained by simpler mechanisms. These may look like harmonicity tuning for a specific stimulus, but may not generalize to other stimuli. For example, a neuron may be a multipeak neuron (identified in many previous studies), tuned to one low and one high frequency tone. Playing both tones together leads to supralinear summation and a significantly higher firing rate. Can your observations be explained by such neurons or is something else required? Are neurons responding to all the harmonics in your stimulus, or just two tones? Assuming it is only two tones, does the frequency separation between these two tones matter? A more parsimonious interpretation is that A2 neurons are integrating multiple frequency components, with perhaps a subset of these neurons are tuned to harmonically related inputs. While the noise correlation data suggest that harmonic stimuli may be processed differently than inharmonic stimuli, many A2 neurons don’t seem to differentiate (Extended data figure 3). And any task or stimulus requiring wideband integration, not just detecting the synchronous onset of harmonics, may benefit from the response properties of A2 neurons. From the data shown in Fig. 2, while A2 neurons have a higher coincidence preference than A1 neurons, many neurons still have no coincidence preference (or even a negative preference), suggesting that this region is more heterogenous than just being the “a locus for harmonic integration” as mentioned in the abstract. To be clear, I am not suggesting any additional experiments, rather a suggestion to tone down your main claims. It is my opinion that it would be more reasonable to say that A2 is involved in processing sounds over a wider spectral and temporal integration window than A1, which is likely advantageous for harmonic and vocalization processing.

We thank this reviewer for their careful consideration of alternative possibilities for explaining our observations. Consistent with this reviewer’s suggestion, we and other previous studies showed that A2 neurons overall are more broadly tuned to pure tones than A1 neurons (Extended Data Fig. 5b). Furthermore, we now include our new analysis showing that there is a

tendency for broader tuning of harmonics-responding neurons (either in the coincidence- or shift-preferring clusters) than harmonics-nonresponsive neurons (Extended Data Fig. 5c). These data support that broad tuning of A2 neurons could facilitate their integration for multi-frequency sounds. However, we would like to note that frequency tuning properties alone do not explain the spectro-temporal integration that we found in A2, as the coincidence-preferring cluster does not have broader tuning than the shift-preferring cluster. To address this in the manuscript, we have updated the Results section to state:

“Although harmonics-responding neurons tended to have broader pure tone tuning than harmonics-nonresponding neurons, we observed no difference between the tuning broadness of coincidence-preferring and shift-preferring neurons (Extended Data Fig. 5c), suggesting that coincidence-preference is not simply conferred by the neurons’ tuning properties.”

We also agree with this reviewer that describing A2 as a locus of harmonic integration is likely an oversimplification, considering its heterogeneity in response properties and its preferential representation of coincident inharmonic sounds. To address this, we have updated our wording throughout the manuscript to tone down our use of “harmonics” and generalized our description more to multi-frequency sounds wherever possible. For example, in the Abstract, we changed the last sentence from “we propose A2 as a locus for harmonic integration” to “we propose A2 as a locus for multi-frequency integration.” We also updated one of the section headings in the Results and Fig. 6 title from “A2 subnetworks exist only for sounds with harmonic regularity” to “A2 subnetworks are preferentially recruited by sounds with harmonic regularity.” Furthermore, we now explicitly mention heterogeneity of A2 in the Discussion by stating,

“At the same time, our data also showed that only 20-40% of A2 neurons displayed coincidence-preference for each frequency combination (Fig. 5), suggesting heterogeneity even within this restricted area. Therefore, future work is needed to examine whether different types of complex sounds are encoded in overlapped or segregated neural populations across these areas. One interesting possibility is that the broad and heterogeneous tuning properties of A2 neurons give them the ability to integrate sounds over a broader range of spectral and temporal space than A1 neurons. AuV-TeA area may possess subnetworks that extract distinct spectro-temporal structures, including harmonics, sweeps, and vocalizations, and serve as a gateway for associating this information with behavioral relevance, analogous to the “ventral auditory stream” in humans and non-human primates.”

Minor Comments:

1. figure 7f-it is difficult to see the small changes between the observed and linear sum. I would recommend having an inset plot for one trace, where you make your point more clearly. Looking at the individual plots, it does seem that some responses are not linear early on, when just the single tone is played, but how can this be the case if the other two tones haven't been played yet...

We thank the reviewer for this comment and have added insets for Figure 7f to better visualize

the differences between the traces and linear sum.

We have checked the traces carefully, but we did not see large nonlinearity for the single tone-leading parts other than the fluctuations at a noise level. It may be possible that this reviewer is referring to the traces on the right-most column, where two tones are leading. If it is the case, this is expected since there is already a sublinear summation even between just two tones.

2. Fig 7: It is hard to interpret the result that the summation is always sublinear, but just less so in A2 (so more linear)...Is this just the consequence of SOM neurons making the response sublinear?

This is an interesting point. Overall sublinearity in the cortical summation of simultaneously presented stimuli has been widely observed across sensory modalities, and multiple mechanisms have been proposed to explain it. For example, one popular circuit-level mechanism is normalization (Carandini and Heeger, Nat Rev Neurosci 2011 doi: 10.1038/nrn3136; Rubin, Van Hooser, and Miller, Neuron 2015 doi: 10.1016/j.neuron.2014.12.026), where increased stimulus intensity recruits balanced inhibition to dampen both the neural firing and excitatory synaptic inputs. Another possibility is, as this reviewer pointed out, that SOM neurons could cause sublinearity in synaptic currents through the delayed suppression of recurrent excitation (Kato et al. Neuron 2017 doi: 10.1016/j.neuron.2017.06.019; Adesnik, Neuron 2017 doi: 10.1016/j.neuron.2017.08.014). Our optogenetic manipulation supports the latter explanation, but we expect that the former mechanism also contributes. We hypothesize that A1 shows dominant sublinearity due to these inhibitory mechanisms. In contrast, the less-sublinear summation in A2 could result from the balance between the inhibitory mechanisms and the preferential recruitment of excitatory networks by coincident multi-frequency sounds.

3. Line 305-“ these results indicate that A2 is causally related to perceptual discrimination of harmonic sounds...”

The behavioural data does not demonstrate the importance of harmonic sounds as harmonicity is not varied. Furthermore, multiple features are changing between stimuli (frequencies static or dynamic, sound level differences, duration, etc.) and no control experiments have been run to demonstrate what acoustic features the animal is using. This is perhaps the weakest part of an otherwise very strong manuscript. I would suggest leading with the behavioural experiment, which is at best suggestive that A2 is doing something different than A1, akin to wideband spectral processing, and then following up with understanding the properties of A2 (harmonic vs. inharmonic), and only then mechanistic issues such as the role of inhibition.

We thank the reviewer for the suggestions. We amended the sentence to,

“Taken together, these results indicate that A2 is causally related to perceptual discrimination of spectro-temporal structures, solidifying its role in coincidence-dependent integration of multi-frequency sounds.”

In addition, to clearly state the uncertainty on the strategy that the animals (and A2 circuits) employ to perform this task, we added the following sentences in the Discussion:

“It is important to note that the involvement of A2 in our specific behavioral task does not necessarily require its specialization for processing harmonicity. A2 could have contributed to this task using other sensory cues, such as the bandwidth of frequency distribution, dynamics in the spectral composition, or changes in sound intensity. Behavior experiments with additional controls for sensory stimuli will be needed to elucidate the range of spectro-temporal features that A2 circuits integrate.”

We also thank the reviewer for the suggestion on the order of the figures. We totally agree with this reviewer that the strongest and most critical parts of our manuscript are the demonstration of inter-area difference (Fig. 1), coincidence-preference in A2 (Fig. 2), and the role of inhibition (Fig. 3-4). However, for this exact reason, we prefer showing these critical data earlier in the manuscript. This is likely a matter of preference, but for us, as circuit neuroscientists rather than behavioral neuroscientists, it seems natural to show the critical circuit dissection data first and then provide supportive behavioral data later.

4.A note on extended data figure 8- positive onsets may have not worked due to combination tones produced at F0 by 2F0 and 3F0 played together (masking the sudden onset of F0 when delayed, as the sound was already present as a combination tone)

We thank the reviewer for pointing this out. We also think that combination tones could be one reason for this temporal asymmetry in the behavior. To include this idea, we revised the Results section to:

“However, since they failed to perform the task for positive Δ onsets, potentially due to masking of the delayed F0 by the generation of a combination tone from the upper components, we focused on negative Δ onsets for the following experiments.”

Reviewer 3

We are delighted that this reviewer kindly stated that *“Overall, the study is timely, interesting and well-executed. The experiments are expertly performed and the manuscript is well written. Comparing and contrasting neural activity and perturbations in A2 vs. A1 is a major strength of the study. I support the publication of this manuscript in Nature Communications.”* Please see our responses to this reviewer below.

Major Comments:

1. Using PCA, Euclidean distance and correlation measures in Figure 2g-i, the authors claim that A1 neuronal representation changes continuously with time onset, consistent with a gain-control model, whereas A2 representation shows more abrupt transitions. However, using NMF in 5a, the authors find three distinct clusters in both A1 and A2. It is unclear why that is so. Authors should clarify the reason for this discrepancy. I am convinced that there are more neurons in A2 for coincidence detection compared to A1 (Figure 5b) but the evidence for gain control in A1 vs. non-overlapping representation in A2 is indirect at best. It would help to provide more direct analyses to convincingly prove this point. For example, does scaling (consistent with gain control) the response profiles of individual neurons in A1 for zero delta onset (for e.g., in Fig 2c) adequately describe the responses at other delta onsets?

We thank the reviewer for pointing out this potentially confusing description in our manuscript. This reviewer raised a concern that the NMF clustering in Fig. 5 is inconsistent with the gain control mechanism in A1. Indeed, we did not mean to propose that the gain control mechanism explains A1 harmonic representations. Most likely, this paragraph triggered confusion because we started it with two alternative hypotheses of gain control and distinct ensemble patterns, whereas we ended it without stating a clear conclusion on A1. In fact, our correlation coefficient data (Fig. 2i) showed that the ensemble patterns changed continuously across Δ onsets in A1 (as opposed to the abrupt transitions we observed in A2). Therefore, even in A1, sounds with different Δ onsets triggered distinct ensemble patterns, and thus, gain control is not at play. We have amended the Results section to be more explicit in our rejection of the gain control hypothesis, stating:

“A1 ensemble activity patterns also varied with Δ onsets, but rather the patterns gradually changed across sound conditions. Thus, harmonics with a range of Δ onsets activate distinct ensemble patterns in both A1 and A2, ruling out the gain-control hypothesis. However, only A2 shows a discrete activity pattern for coincident harmonics, suggesting a link between its activity and the perceptual binding of simultaneous sound components.”

Since A1 showed distinct ensemble patterns across Δ onsets, the presence of coincidence-preferring neurons in Fig. 5 is not surprising. The fraction of these neurons was not enough to drive ensemble patterns away from those of temporally shifted harmonics. This is in contrast to A2, where a large number of coincidence-preferring neurons strongly pushed the ensemble representations away from those of shifted harmonics, causing abrupt transitions. To express this idea, we added a sentence in the Results stating:

“This overrepresentation of coincidence-preferring cells in A2 likely explains the abrupt transitions of ensemble representation around Δ onset = 0 in this area while those in A1 transitioned more continuously.”

We hope that our text changes more accurately explain the differences in ensemble representation we observed between A1 and A2.

2. Inferring functional subnetworks using noise correlation analyses in Figure 5f-g and Figure 6i, is the weakest part of the paper. Noise correlation analysis is meant to capture the correlated fluctuations of neuronal firing rates between simultaneously recorded neurons and is, at best, an indirect surrogate for ground-truth connectivity. I have several methodological concerns about this analysis. First, estimating noise correlations using calcium imaging is especially challenging because of temporal smoothing of spiking activity that obscures the co-fluctuations that one is trying to measure.

We are wondering if this reviewer is referring to a noise correlation measurement method that calculates the Pearson correlation between the temporal fluctuations in the firing of a pair of neurons. Instead, in our analysis, we calculated noise correlation as the Pearson correlation between the trial-to-trial fluctuations of the mean-subtracted sound-evoked responses of pairs of simultaneously recorded neurons. In the latter method, Pearson correlation is calculated using the total spike count triggered by a stimulus in each trial, and therefore temporal smoothing does not significantly influence the calculation. Noise correlation measured similarly has been popularly used in two-photon calcium imaging experiments (Rothschild, Nelken, and Mizrahi, *Nat Neurosci* 2010, doi: 10.1038/nn.2484; Kanold lab, *Neuron* 2018, doi: 10.1016/j.neuron.2018.01.019). Moreover, recent work combining simultaneous two-photon calcium imaging and single cell-targeted optogenetics demonstrated that the noise correlation between a pair of neurons reflected their functional influence on each other, supporting the idea that noise correlation measured with calcium imaging reflects local connectivity (Chettih and Harvey, *Nature* 2019, doi: 10.1038/s41586-019-0997-6). Therefore, we believe that our methodology is accepted in the field. Nevertheless, we agree with the reviewer that noise correlation is still an indirect measure, and high noise correlation does not necessarily indicate direct connectivity. To indicate this caution in the interpretation, we rewrote Discussion to state:

“Although an indirect measure, high noise correlation between neurons suggests that they could receive shared presynaptic inputs, form mutual connections, and/or indirectly interact through multi-synaptic pathways. Interestingly, whether direct or indirect, these connectivity schemes...”

Second, it is unclear from the method section whether this analysis was done on simultaneously recorded neurons (which should be the case) or on a pseudopopulation of concatenated neurons across multiple sessions.

We apologize for the lack of clarity on this point. Noise correlation was only measured between pairs of simultaneously imaged neurons. We have added a statement to the Methods:

“Noise correlation was calculated between simultaneously recorded ROIs in each mouse, and the values were concatenated across mice.”

We additionally updated the Results section to read,

“To test this, we asked whether coincident harmonics-preferring neurons in A2 form functional subnetworks by measuring the pairwise noise correlation between simultaneously imaged neurons during the presentations of harmonic stimuli.”

Third, noise correlations are very sensitive to the SNR of the calcium traces and adequate controls need to be performed. Fourth, the difference in noise correlations between A1 and A2 is statistically significant (Figure 5g) but too subtle to allay my concerns.

We thank the reviewer for suggesting this important control. To address this, we now performed additional analyses in which we controlled for the noise level (Extended Data Fig. 4c, 6f) and the signal-to-noise ratio (Extended Data Figure 4d, 6g) of individual neurons. Here, we calculated the noise level of each cell as the standard deviation (Std) of the dF/F trace during the baseline without stimuli. Signal-to-noise ratio (SNR) of each cell was calculated as the sound-evoked response to its most preferred stimulus divided by Std. For each cell pair in the Same-F0 cluster, noise correlation was calculated between the Std-matched or SNR-matched random cell pairs drawn from the same field of view. We found essentially identical results in both control analyses as our original distance-matched control: Same-F0 cell pairs in A2 had significantly higher noise correlation than either Std-matched or SNR-matched controls ($p < 0.0001$ in both cases), suggesting that our observations are not due to noise. We now edited the Results and Extended Data Fig. 4 stating,

“Differences in noise levels or signal-to-noise ratios across cells did not account for the higher noise correlation observed within the same-F0 cluster (Extended Data Fig. 4c, d). These data suggest the existence of F0-specific functional subnetworks of coincidence-prefering neurons in A2, which is not simply a consequence of their spatial proximity or activity level.”

and amended the Methods section to read,

“To account for the dependence of noise correlation on imaging sensitivity, two separate controls were taken: noise level and signal-to-noise ratio. Noise level of each ROI was calculated as the standard deviation (Std) of dF/F traces during the baseline period without sound stimuli. The signal to noise ratio (SNR) of each ROI was calculated as the peak sound-evoked response amplitude divided by Std. For each ROI, two random ROIs with the closest Std (or SNR) were drawn from the same field of view. Thus, each Same-F0 cell pair had four Std (or SNR)-matched control cell pairs: combinations between 2 x 2 control ROI sets.”

We agree with the reviewer that the noise correlation data for A1 is not as strong as that in A2 regarding its statistical power, since A1 does not have many coincidence-prefering neurons (consistent with our overall conclusion). This is why we did not explicitly describe the noise correlation results for A1 in our Results section, although we still included it in the figures since many readers would be curious to see the results in both areas. We believe that the critical result here is the comparison between our observed A2 data with its own controls. We hope that the newly added Std-matched and SNR-matched controls, in addition to our original distance-matched control, have further solidified our findings.

Importantly, the major results of the paper about A2 neural activity, circuit mechanism and behavioral role still hold without claiming that “Therefore, while A2 integrates coincident sounds regardless of spectral structure, it has specialized excitatory subnetworks dedicated to binding harmonics.” In my opinion, the sections on inferring excitatory subnetworks using noise correlations distracts from the main results.

We agree with this reviewer that the strongest and most critical parts of our manuscript are the demonstration of inter-area difference, coincidence-preference in A2, and the role of inhibition. These data, together with the behavior results, convey our key findings on the contribution of A2 to the integration of coincident multi-frequency sounds. However, while SOM cell-mediated inhibition can explain coincidence-preference, inhibitory mechanisms alone are not sufficient in explaining the preferential integration of multi-frequency sounds in A2. The data for noise correlation analyses and whole-cell recording experiments, both of which suggesting the contribution of excitatory mechanisms, are supportive additions in this respect. However, we admit that we may have overstated these conclusions in the manuscript, and therefore we removed the sentence from the Results section that stated, “Therefore, while A2 integrates coincident sounds regardless of spectral structure, it has specialized excitatory subnetworks dedicated to binding harmonics.” In addition, we have updated the wording throughout the manuscript to avoid overstatements on the harmonics-specific subnetworks based on the noise correlation analysis, as we already stated in our responses above.

We hope all these changes regarding noise correlation improved the clarity of our manuscript and helped avoiding unnecessary distraction from our main conclusions.

3. A major result of this study is the contribution of A2 (but not A1) in perceptual discrimination of harmonics (Figure 8). Are the psychometric curves plotted in Figure 8g from an example animal? Was there any effect on reaction times after A1 and A2 inactivation? Authors should also include a supplementary figure panel to show the individual psychometric curves for each animal to give a sense of behavioral variability and the effect of optogenetic silencing, in addition to the half-max delta onset summary in Figure 8h. Instead of focusing on a point estimate of delta onset at half-max performance, the authors could try to include all data points and quantify the slope of the individual response curves. The expectation is that the distribution of slopes will be shallower for A2 inactivation compared to control.

The psychometric curves in Fig. 8g are from representative animals, as stated in the figure legend. We now included a supplemental figure showing the performance of each mouse with A1 or A2 inactivation (Extended Data Fig. 10). As can be seen in the data of individual mice, the psychometric lines showed curved shapes, and thus were poorly fit by straight lines. Therefore, we instead took the logistic growth rate as the “slope” value and included the data in the new Fig. 8i. We found that there is no significant difference between the slopes in control and LED trials in either A1 or A2. This was also the case when we used straight fit lines including all data points. This result suggests that the A2 inactivation shifts the entire psychometric curve to the

longer Δ onset values rather than changing the shape of the curve. We replaced the representative A2 psychometric curves in Fig. 8g to better represent our results.

We thank this reviewer for the suggestion on the lick latency analysis. However, unfortunately, our task design does not allow us to measure reaction time in a meaningful way. Our experiment employs a sound offset-lick task, where the answer period starts after the sound offset. The mice are allowed to lick during the sound, but they do not receive rewards until the designated answer period after the sound offset. Therefore, our mice typically learn to lick only after the sound offsets, and thus the timing of lick is independent of the perceptual decision that happens around the sound onsets. We nevertheless calculated the lick latency (measured as the first lick after sound offset) for the hit trials and showed the data in the figure below. We found no significant difference between lick latency in control and LED trials in either A1 or A2 inactivation. This result demonstrates the intact ability of mice to react to sound cues during optogenetic manipulation, but it likely does not reflect the perceptual discrimination speed.

Fig. Latency to first lick during the answer period for A1 and A2 mice between control and LED trials. A1: ctrl: 0.090 ± 0.01 , LED: 0.087 ± 0.01 , $p = 0.359$; A2: ctrl: 0.064 ± 0.01 , LED: 0.065 ± 0.01 , $p = 1.000$.

Minor comments:

1. Can the authors provide some justification for the parameter choices (highlighted below) for the 2-photon imaging response calculation pipeline? This is important given that all subsequent analyses depend upon this step. What fraction of neurons end up being significantly modulated based on these choices? Please include this number either in the main text or in the figure 2 legend.

The remaining pixels were averaged to create a background fluorescence time series $F_{background}(t)$. The fluorescence signal of a cell body was estimated as $F(t) = F_{cell_measured}(t) - 0.9 \times F_{background}(t)$. To ensure robust neuropil subtraction, only cell ROIs that were at least 3% brighter than the background ROIs were included. Normalized time series dF/F were generated after a small offset (20 a.u.) was added to $F(t)$ in order to avoid division by extremely low baseline values in rare cases. Harmonics-evoked responses were measured during 1-s window after sound onsets. Cells were judged as significantly excited if they fulfilled two criteria: 1) dF/F had to exceed a fixed threshold value consecutively for at least 0.5 s in more than half of trials. 2) dF/F averaged across trials had to exceed a fixed threshold value consecutively for at least 0.5 s."

Appropriate background subtraction factor depends on the optics and the clarity of tissues, so we empirically chose 0.9 based on the comparisons between our original movies and the resulting dF/F traces. Our value is within the range that have been typically adopted by labs with similar imaging systems (1.0 in Komiyama lab, Nature 2014 doi: 10.1038/nature13235; 0.7 in Svoboda lab, Nature 2013 doi: 10.1038/nature12354). We note that the way we define “background pixels” more stringently excludes the dendrites from surrounding neurons than the automated systems like Suite2P, and that might also affect the appropriate value for the subtraction factor.

The rules of “at least 3% brighter than the background” and “addition of a small offset (20 a.u.)” were added in order to avoid the near-zero denominator in dF/F calculations. In GCaMP imaging, some cells are visible only when they flash. In these cells, background subtraction could bring the baseline fluorescence to zero, resulting in unreasonably large dF/F values. These two rules help prevent the skewing of the population data by avoiding the generation of incorrect dF/F values.

The 1-s time window for response measurement was determined considering the slow kinetics of GCaMP6s. The two criteria for determining significant responses were meant for avoiding two types of noises. 1) We chose to require supra-threshold activity in more than half of trials to only detect reproducible responses as significant. If we rely only on the averaged traces across trials, it could lead to the false detection of cells that happened to give a huge spontaneous fluctuation in only one trial. Simple use of statistics like t-test also suffers from this type of noise. 2) We chose to require the averaged traces to exceed a threshold consecutively for at least 0.5 s in order to avoid the false detection of cells with high noise in dF/F traces. Cells with high noise could reach the 3.3×SD threshold (which we determined by receiver operator characteristic analysis) randomly in some frames just by chance. Setting a duration threshold ensures that the detected responses are not artifacts due to random brief noise.

We now included the fraction of significantly responsive neurons in Methods as:

“Overall, the fraction of pyramidal cells with significant increases in the activity to at least one harmonic stimuli was 33% (n = 336 out of 1109 cells) in A1 and 42% (n = 297 out of 713 cells) in A2.”

2. Single unit example in Fig 4b is not very convincing because the control responses (black traces) does not decay much at non-zero delta onsets, making it hard to see the difference after SOM inactivation.

We appreciate this reviewer bringing up ways to improve our figures. We believe that the difficulty in seeing the optogenetic effect is because SOM inactivation did not change the peak amplitude of the traces as much as it extended the response durations. To better visualize this elongation of responses during SOM inactivation, we scaled up the x axis of Figures 4b and 4g. We hope this change makes the optogenetic effects more evident to the eyes.

3. Collapsing across all delta onsets to plot the cumulative distributions in Fig 4m and 4p obscures the effect, which is predominant at small deviations from 0 (Fig 4o).

Actually, coincidence-preference index (CI) was calculated by comparing the response to the coincident harmonics ($\Delta\text{onset} = 0$) and the response to harmonics with small Δonsets (+/-15 and +/-30 ms, pooled). Therefore, we believe our calculation is exactly what this reviewer suggested. We chose to include +/-15 and +/-30 ms (and not +/-45 ms) since these two temporal shifts seemed to show the clearest contrast to coincident sounds in pyramidal cell imaging data. We used the same condition for calculating CI throughout the manuscript for consistency, including pyramidal cell calcium imaging, inhibitory neuron imaging, and unit recording.

4. On page 9 first paragraph, the authors suggest that “similar to pyramidal cells, PV cells exhibited strong coincidence preference in A2, but less in A1 (Fig. 4o, p; A1: 0.066 ± 0.003 ; A2: 0.171 ± 0.006 ; A1 vs A2: $p = 0.0001$), suggesting that they follow the response properties of surrounding pyramidal cells.” Given these experiments, it is impossible to know if the PV neurons follow the pyramidal cells or vice versa.

This reviewer is correct here in that we cannot assume the causal relationship. Therefore, we amended the Results section to:

“Similar to pyramidal cells, PV cells exhibited strong coincidence preference in A2, but less in A1, suggesting their co-modulation with surrounding pyramidal cells.”

5. Figure 8e should include electrophysiology data examples from A2 in addition to A1, since that is the focus of the study.

We have updated Fig. 8e and 8f to show electrophysiology data examples for both A1 and A2.

REVIEWER COMMENTS

Reviewer #2 (Remarks to the Author):

the revised manuscript looks great. Nice job! Happy to support publication.

Reviewer #3 (Remarks to the Author):

The authors have adequately addressed my concerns. No additional issues. I congratulate the authors on this exciting study.